# BABILong: Testing the Limits of LLMs with Long Context Reasoning-in-a-Haystack

**Yuri Kuratov**[* 1,2]    **Aydar Bulatov**[* 2]    **Petr Anokhin**[1]    **Ivan Rodkin**[2]
**Dmitry Sorokin**[1]    **Artyom Sorokin**[1]    **Mikhail Burtsev**[3]

[1]AIRI, Moscow, Russia   [2]Neural Networks and Deep Learning Lab, MIPT, Dolgoprudny, Russia
[3]London Institute for Mathematical Sciences, London, UK
{yurii.kuratov,bulatov.as}@phystech.edu, mb@lims.ac.uk

## Abstract

In recent years, the input context sizes of large language models (LLMs) have increased dramatically. However, existing evaluation methods have not kept pace, failing to comprehensively assess the efficiency of models in handling long contexts. To bridge this gap, we introduce the BABILong benchmark, designed to test language models' ability to reason across facts distributed in extremely long documents. BABILong includes a diverse set of 20 reasoning tasks, including fact chaining, simple induction, deduction, counting, and handling lists/sets. These tasks are challenging on their own, and even more demanding when the required facts are scattered across long natural text. Our evaluations show that popular LLMs effectively utilize only 10-20% of the context and their performance declines sharply with increased reasoning complexity. Among alternatives to in-context reasoning, Retrieval-Augmented Generation methods achieve a modest 60% accuracy on single-fact question answering, independent of context length. Among context extension methods, the highest performance is demonstrated by recurrent memory transformers after fine-tuning, enabling the processing of lengths up to 50 million tokens. The BABILong benchmark is extendable to any length to support the evaluation of new upcoming models with increased capabilities, and we provide splits up to 10 million token lengths.

## 1   Introduction

Today, large language models (LLMs) and neural architectures are continually evolving and achieving remarkable improvements, particularly in their ability to handle longer contexts (OpenAI, 2023b; Reid et al., 2024; Anthropic, 2024). The ability of these models to process and generate text based on rich contextual information is crucial for several reasons. For example, longer contexts provide more information for the model to condition its outputs, leading to more accurate, contextually relevant, and up-to-date responses. Furthermore, long-context capabilities can enhance in-context learning by providing more in-context examples, instructions to follow, or example trajectories in context of reinforcement learning (Chevalier et al., 2023; Agarwal et al., 2024; Lee et al., 2024).

Despite these advances in models capabilities, the benchmarks used to evaluate them have not kept pace. For example, current benchmarks such as Longbench (Bai et al., 2023) and L-Eval (An et al., 2023) scale only up to 40,000 tokens, while models are capable of hundreds of thousands and millions of tokens (Rodkin et al., 2024; Reid et al., 2024; Bulatov et al., 2024; Anthropic, 2024; Liu et al., 2024a; Gu & Dao, 2023; OpenAI, 2023a).

Creating natural and comprehensive long-context benchmarks that are human labeled is very challenging. As a consequence, synthetic benchmarks focusing on variations of "needle-in-a-haystack"

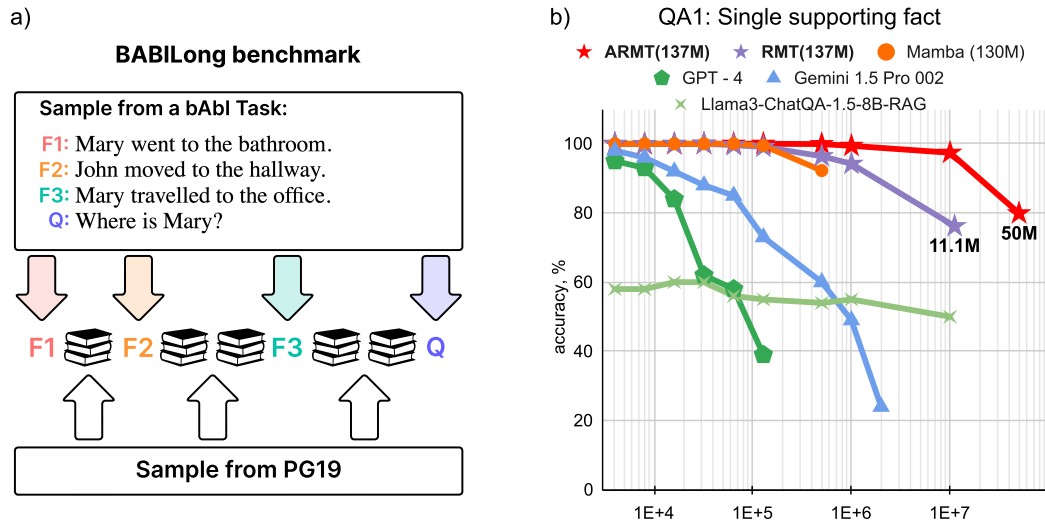

Figure 1: **a) Generation of BABILong dataset.** Facts relevant for the question are hidden inside a larger background texts from PG19. **b) Recurrent transformers answer questions about facts from very long texts when retrieval augmented generation fails.** Common RAG method fails to answer questions because order of facts matters. GPT-4 effectively uses only about 10% of the full 128K window. Gemini 1.5 Pro shows strong performance up to 64K tokens. Small LMs, ARMT & RMT with GPT-2 (137M) and Mamba (130M) fine-tuned for the task are able to solve it, with recurrent memory transformers scoring well up to record 50 000 000 tokens. Here we show the best results obtained by models.

tasks have become increasingly common (Zhang et al., 2024b; Liu et al., 2024a; Song et al., 2024b; Hsieh et al., 2024). One widely used needle-in-a-haystack task involves finding specific "needles with magic numbers" in a haystack of Paul Graham's essays[1]. However, the widespread use of this approach has highlighted its limitations - it is overly simplistic, and novel long context models often achieve perfect performance, as usually demonstrated by fully green heatmaps (Reid et al., 2024; Cohere, 2024; Liu et al., 2024a; Wang et al., 2024c). This shows that while it serves well as a basic verification tool, it is not a rigorous benchmark that can effectively challenge and differentiate advanced long-context models. Another major drawback of the original default setup[1] is that model predictions are evaluated and scored by an LLM (GPT-3.5-turbo) on a scale of 1 to 10, with the same single needle used for each position and document length. While averaging over multiple different needles can provide more robust results.

To bridge this gap, we introduce the BABILong benchmark, designed to test language models' ability to reason across facts distributed in extremely long documents. BABILong includes a diverse set of 20 reasoning tasks, including fact chaining, simple induction, deduction, counting, and handling lists/sets, that were designed as prerequisites for any system that aims to be capable of conversing with a human (Weston et al., 2016). As a source of long natural documents we use books from PG19 corpora (Rae et al., 2020). In this way, BABILong allows the construction of tasks of almost arbitrary length, in order to adapt them to the evaluation of new, more powerful models in an extensible and controllable way. We provide sets of predefined lengths with splits up to 10 million tokens, and we evaluate models on samples with up to 50 million tokens.

We find that popular LLMs effectively use only 10-20% of the context, with performance declining sharply as length and task complexity increase. Retrieval-Augmented Generation methods achieve a modest 60% accuracy in answering single-fact questions, regardless of context length. Among other methods, Mamba and Recurrent Memory Transformers (RMT and ARMT) show the highest performance, with ARMT capable of processing lengths up to 50 million tokens.

The main contributions of our work are as follows:

1. We introduce BABILong, a novel scalable generative multi-task benchmark for evaluating the performance of NLP models in processing arbitrarily long documents with distributed facts.

2. We evaluate over 30 recent long-input language models with various sizes, architectures, and context extension methods on BABILong.

---

[1]https://github.com/gkamradt/LLMTest_NeedleInAHaystack

Table 1: **The first ten tasks of BABILong with the number of supporting and distracting facts.** The last column displays the performance of LLMs on each task in the absence of background text. Each dot represents one of the selected models, while the blue bars indicate the median accuracy across tested models.

| TASK | NAME | FACTS PER TASK | RELEVANT FACTS PER TASK | LLMs ANSWER ACCURACY WITHOUT BACKGROUND TEXT (0K) |
|------|------|----------------|--------------------------|---------------------------------------------------|
| QA1 | SINGLE SUPPORTING FACT | 2-10 | 1 | 99 |
| QA2 | TWO SUPPORTING FACTS | 2-68 | 2 | 64 |
| QA3 | THREE SUPPORTING FACTS | 4-320 | 3 | 38 |
| QA4 | TWO ARG RELATIONS | 2 | 1 | 55 |
| QA5 | THREE ARG RELATIONS | 2-126 | 1 | 80 |
| QA6 | YES-NO QUESTIONS | 2-26 | 1 | 91 |
| QA7 | COUNTING | 2-52 | 1-10 | 28 |
| QA8 | LISTS-SETS | 2-50 | 1-8 | 77 |
| QA9 | SIMPLE NEGATION | 2-10 | 1 | 89 |
| QA10 | INDEFINITE KNOWLEDGE | 2-10 | 1 | 80 |

3. We find that popular LLMs effectively utilize only 10-20% of the context, with performance degrading sharply as reasoning complexity increases. Retrieval augmented generation fails to demonstrate good scores but fine-tuning for specific task helps.

4. We demonstrate successful in domain single fact question answering with the recurrent memory transformer on input texts up to 50 million tokens, which is a record for the sequence size processed by a single model.

The BABILong benchmark data and code for evaluation are available.[2]

## 2   The BABILong Benchmark for Long Context Processing

The fundamental concept behind the **B**enchmark for **A**rtificial **I**ntelligence for **Long**-context evaluation is extending the length of existing tasks to test the ability of generative models in handling long contexts. Solving tasks with a long context size requires the model to distinguish important information from large amounts of irrelevant details. To simulate this behavior we "hide" the sentences of the original task between the sentences of irrelevant text that is drawn from another closely related distribution (see Figure 1a). Examples are constructed by gradually adding new sentences from the background dataset in their natural order until the augmented sample reaches the desired length. This way, we are not bound by the length of the original task itself, making it possible to assess even the longest available models with context sizes up to millions of tokens. For background text we use books from the PG19 dataset (Rae et al., 2020) due to the substantial book lengths and naturally occurring long contexts. The model is required first to distinguish the sentences related to the original task, then memorize and subsequently utilize them to generate the correct solution.

In this work we extend the bAbI benchmark (Weston et al., 2016), which consists of 20 tasks designed to evaluate basic aspects of reasoning. These tasks are generated by simulating interactions among characters and objects across various locations, each represented as a fact, such as "Mary traveled to the office." The challenge is to answer questions based on the facts generated in the current simulation, such as "Where is Mary?" The tasks in bAbI vary in the number of facts, question complexity, and the reasoning skills they assess, including spatial and temporal reasoning, deduction, and coreference resolution. In our paper, we label these tasks from 'QA1' to 'QA20'. The first ten tasks, as shown in Table 1 demonstrate that current LLMs exhibit mixed performance even without distractor texts, indicating that the BABILong tasks span a broad spectrum of difficulty and allow for testing models across various performance dimensions. Details and performance metrics for the bAbI tasks, along with examples of BABILong samples generated using our pipeline, can be found in Appendix L.

As evident in the following sections, these seemingly simple tasks pose significant challenges to language models. Although filtering facts from background text might be straightforward, models encounter next challenges of finding supporting facts among distractors and performing types of reasoning such as counting that are especially difficult for LLMs. Additionally, most NLP benchmarks

---

[2]code: `https://github.com/booydar/babilong`, evaluation dataset: `https://huggingface.co/datasets/RMT-team/babilong`, leaderboard: `https://huggingface.co/spaces/RMT-team/babilong`

Table 2: **BABILong is a challenging benchmark for current long-context models.** Even models that claim to support 128K tokens experience degradation beyond 10% of their input capacity. RAG methods do not help, while fine-tuning of small scale models (ARMT and RMT, 137M and Mamba, 130M) shows that the tasks are solvable. Values represent average accuracy over QA1-QA5 tasks from BABILong. Models are grouped by the length of the context they claim to support.

| Input size | GPT-2 (137M) | mamba-2.8b-hf | rwkv-6-world-7b | v5-Eagle-7B-HF | gemma-2-9b-it | Meta-Llama-3-8B-Instruct | recurrentgemma-9b-it | LLaMA-2-7B-32K | longchat-7b-v1.5-32k | LongAlpaca-13B | Llama-2-7B-32k-Instruct | 01-ai/Yi-34B | Mistral-7b-Instruct-v0.3 | Mistral-7b-Instruct-v0.2 | Mixtral-8x7B-Instruct-v0.1 | 01-ai/Yi-34B-200k | Mixtral-8x22B-Instruct-v0.1 | activation-beacon-llama2-7b-chat | Yarn-Mistral-7b-128k | chatglm3-6b-128k | activation-beacon-mistral-7b | 01-ai/Yi-9B-200k | Phi-3-mini-128k-instruct (3.8B) | ai21labs/Jamba-v0.1 (12B/52B) | Phi-3.5-mini-instruct (3.8B) | c4ai-command-r-v01 (35B) | Phi-3.5-MoE-instruct (6.6B/61B) | Phi-3-medium-128k-instruct (14B) | Meta-Llama-3.1-8B-Instruct | gpt-4o-mini-2024-07-18 | Qwen2.5-7B-Instruct | GPT-4 (0125-preview) | Meta-Llama-3.1-70B-Instruct | Qwen2.5-72B-Instruct | – Mamba (130M) fine-tune | Gemini 1.5 Pro 002 | Llama3-ChatQA-1.5-9B + RAG | – RMT (137M) fine-tune | – ARMT (137M) fine-tune | input size |
|---|---|---|---|---|---|---|---|---|---|---|---|---|---|---|---|---|---|---|---|---|---|---|---|---|---|---|---|---|---|---|---|---|---|---|---|---|---|---|---|---|
| 0K | 27 | 70 | 56 | 62 | 72 | 64 | 62 | 41 | 46 | 48 | 49 | 72 | 59 | 60 | 65 | 65 | 75 | 55 | 51 | 56 | 59 | 52 | 64 | 65 | 70 | 64 | 67 | 72 | 67 | 74 | 68 | 87 | 85 | 83 | 98 | 92 | 48 | 99 | 99 | 0K |
| 1K | 15 | 52 | 55 | 54 | 71 | 60 | 59 | 53 | 42 | 47 | 52 | 52 | 60 | 56 | 63 | 59 | 73 | 52 | 52 | 55 | 56 | 55 | 57 | 53 | 70 | 64 | 68 | 70 | 68 | 72 | 68 | 81 | 81 | 78 | 99 | 88 | 48 | 97 | 96 | 1K |
| 2K | 35 |  | 48 | 48 | 69 | 58 | 57 | 45 | 40 | 46 | 49 | 43 | 55 | 52 | 60 | 56 | 70 | 47 | 43 | 51 | 51 | 48 | 55 | 50 | 62 | 63 | 66 | 67 | 66 | 71 | 68 | 77 | 78 | 76 | 99 | 82 | 47 | 95 | 98 | 2K |
| 4K | 9 |  | 35 | 41 | 64 | 50 | 21 | 40 | 41 | 43 | 43 | 37 | 45 | 49 | 55 | 54 | 65 | 43 | 40 | 48 | 48 | 46 | 51 | 48 | 59 | 61 | 66 | 62 | 66 | 65 | 66 | 74 | 74 | 75 | 99 | 78 | 46 | 92 | 98 | 4K |
| **8K** | 0 | 7 | 2 | 0 | 44 | 14 | 39 | 42 | 40 | 40 | 38 | 29 | 45 | 50 | 52 | 58 | 36 | 38 | 46 | 43 | 45 | 45 | 50 | 46 | 58 | 59 | 62 | 60 | 62 | 62 | 65 | 71 | 70 | 73 | 99 | 75 | 45 | 90 | 98 | **8K** |
| 16K |  |  |  |  |  |  | 8 | 32 | 39 | 36 | 35 | 31 | 26 | 42 | 46 | 50 | 51 | 23 | 30 | 41 | 37 | 36 | 46 | 45 | 53 | 52 | 60 | 57 | 60 | 60 | 63 | 64 | 65 | 71 | 99 | 72 | 45 | 86 | 98 | 16K |
| **32K** |  |  |  |  |  |  |  |  | 3 | 5 | 4 | 5 | 4 | 29 | 37 | 40 | 48 | 43 | 16 | 16 | 36 | 37 | 42 | 41 | 43 | 51 | 56 | 53 | 56 | 54 | 60 | 53 | 59 | 68 | 98 | 68 | 44 | 78 | 98 | **32K** |
| 64K |  |  |  |  |  |  |  |  |  |  |  |  |  |  |  |  |  | 48 | 35 | 8 | 10 | 21 | 27 | 29 | 37 | 40 | 38 | 46 | 49 | 45 | 49 | 45 | 53 | 66 | 97 | 64 | 42 | 70 | 98 | 64K |
| **128K** |  |  |  |  |  |  |  |  |  |  |  |  |  |  |  |  |  | 6 | 9 | 13 | 14 | 24 | 7 | 34 | 10 | 38 | 39 | 30 | 39 | 43 | 45 | 36 | 45 | 58 | 93 | 59 | 45 | 59 | 97 | **128K** |
| 512K |  |  |  |  |  |  |  |  |  |  |  |  |  |  |  |  |  |  |  |  |  |  |  |  |  |  |  |  |  |  |  |  |  |  |  | 51 | 42 | 46 | 95 | 512K |
| **1M** |  |  |  |  |  |  |  |  |  |  |  |  |  |  |  |  |  |  |  |  |  |  |  |  |  |  |  |  |  |  |  |  |  |  |  | 42 | 39 | 43 | 93 | **1M** |
| 10M |  |  |  |  |  |  |  |  |  |  |  |  |  |  |  |  |  |  |  |  |  |  |  |  |  |  |  |  |  |  |  |  |  |  |  | 37 |  | 34 | 77 | 10M |
| avg | 21 | 33 | 40 | 41 | 55 | 55 | 37 | 36 | 36 | 38 | 39 | 40 | 43 | 49 | 54 | 54 | 59 | 32 | 32 | 41 | 41 | 41 | 45 | 47 | 51 | 55 | 59 | 58 | 59 | 61 | 62 | 65 | 68 | 72 | 98 | 70 | 44 | 74 | 95 | avg |

are vulnerable to data leakage to training sets of modern large language models (Sainz et al., 2023). Generated benchmarks, such as bAbI and BABILong, are immune to this type of contamination.

In this work we deliberately employ simple algorithmic tasks to underscore the fundamental limitations of current models in collecting evidence over long contexts even for basic reasoning. The brevity and similarity of the task sentences also enable the model distinguish them from seemingly close background text with the assistance of few-shot examples. This difference in distributions enables the scalability of BABILong to large amounts of diverse noise. Nevertheless, the BABILong approach can be applied to incorporate more complex tasks, using the same strategy of mixing task sentences with background text.

# 3 Benchmarking Results

To maximize value for the research community, we have included models with the highest number of monthly downloads[3] from the Hugging Face platform in our evaluation such as LLama-3 (AI@Meta, 2024); 32k-64k – Mistral (Jiang et al., 2023), Mixtral (Jiang et al., 2024); 128k – ChatGLM3 (Du et al., 2022), LLama-3.1 (Dubey et al., 2024), Phi-3 and Phi-3.5 (Abdin et al., 2024), Command-R (Cohere, 2024), Qwen-2.5 (Team, 2024); 200k – Yi (Young et al., 2024); including long-context fine-tuning: LongChat (Li et al., 2023a), LLama-2-7b-32k[4], LongAlpaca (Chen et al., 2023); long-context adaptation methods: Yarnv2 Mistral (Peng et al., 2023b), Mistral and LLama-2 with Activation Beacons (Zhang et al., 2024a). As a reference, we included GPT-4 (gpt-4-0125-preview) and Gemini 1.5 Pro 002, currently the most powerful model available. Retrieval-augmented generation was also tested, as it represents a common solution for long document QA. As alternatives to traditional architectures, we considered the Mamba (Gu & Dao, 2023), Jamba (Lieber et al., 2024), Recurrent Memory Transformer (RMT) (Bulatov et al., 2022), and Associative RMT (ARMT) (Rodkin et al., 2024). Summary of evaluation results is presented in the Table 2. The table reports average accuracy of models on the first 5 tasks (QA1-QA5) of BABILong for different context sizes. For evaluation details for each task see Table 4 and Appendix C.

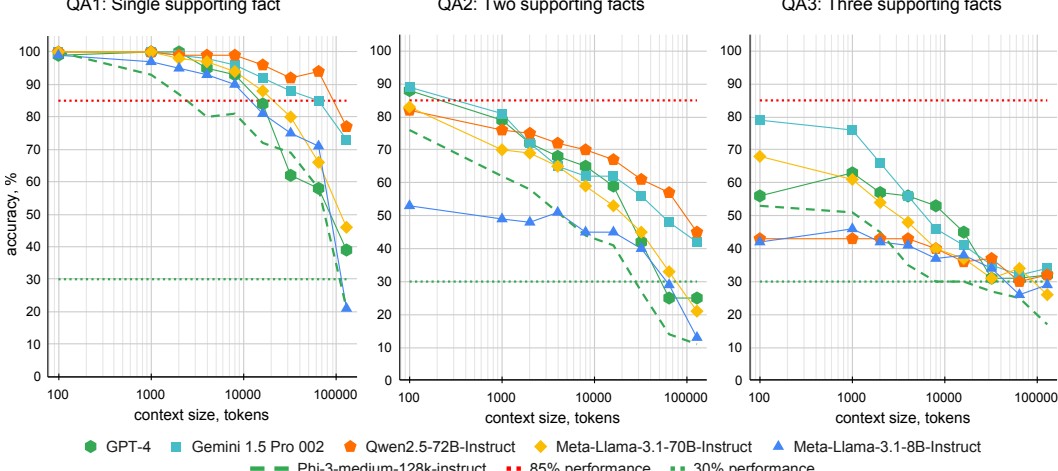

Figure 2: **LLMs struggle to answer questions about facts in texts larger than 10,000 tokens.** The plots demonstrate how the performance of selected leading models deteriorates with increasing context size. For single supporting fact questions (QA1), the majority of models perform well up to 4,000 tokens. However, when a correct response requires two (QA2) or three (QA3) facts, LLMs fail to achieve satisfactory accuracy.

## 3.1 Evaluation of Effective Context Size

One of the most important questions regarding performance of long-context models is how effectively they utilize the input context. Ideally, a model should maintain uniformly high performance regardless of the input size. For instance, if an LLM can process 128K tokens, it is expected to use all of this context in addressing the user's task.

We evaluated the performance of models on question-answering tasks with varying numbers of supporting facts (QA1-QA3) to study how LLMs utilize the available context. Here, we distinguish between a QA task, which requires a single correct answer, and an information retrieval task, which should generate a list of relevant facts or references to information sources. We consider performance satisfactory if the accuracy of an answer exceeds 85% and a complete failure if it is below 30%.[5]

Our benchmarking results show that current LLMs do not efficiently use their full context (Fig. 2). Only 23 out of 34 tested LLMs were able to correctly answer 85% or more of the questions for any of QA1-QA3 tasks in a baseline setting without any background distractor text. Even for the the simplest task involving a single supporting fact (QA1), the majority of models are only able to efficiently use up to 4K tokens, except for GPT-4 and LLama-3.1-70b, which perform well up to 16K, as well as Qwen-2.5-70b and Gemini Pro 1.5 up to 64K. The range of full context utilization on QA1 varies from 5% to maximum 50%. When two supporting facts are required for an answer, only GPT-4 and Gemini Pro 1.5 can solve the task without background text. When facts are embedded within texts, all tested LLMs fall below 85% performance(Fig. 2, QA2). The task with three supporting facts proves to be extremely challenging to current LLMs, with the best scores falling below 80% (Fig. 2, QA3).

Going deeper in performance of specific models presented in the Table 2 we found the following. Yarn fails to extend to longer contexts despite showing stable results in long-context language modeling (Peng et al., 2023b). LongChat, LongAlpaca, and both LLama2-7B-32K and LLama2-7B-32K-instruct models, even when fine-tuned on 32K lengths, failed to perform well on 32K tokens. Activation Beacon performed better than Yarn context extension method for Mistral 7B, but still achieved low results (< 40%) on 32K contexts. In contrast, Mistral-v0.2 and Mixtral-v0.1, trained on lengths up to 32K, performed well on these lengths. Yi-9B-200k, trained on sequences up to 200K, shows less than 30% on 64K tokens and more. Yi-34B-200k shows very promising and stable

---

[3]As of May 2024. Since then we have added new models such as Mistral v0.3, LLama-3.1, Qwen-2.5, Phi-3.5, and GPT-4o-mini.

[4]`https://huggingface.co/togethercomputer/Llama-2-7B-32K-Instruct`

[5]This definition of satisfactory performance is not universal and should be adapted to the specific task at hand.

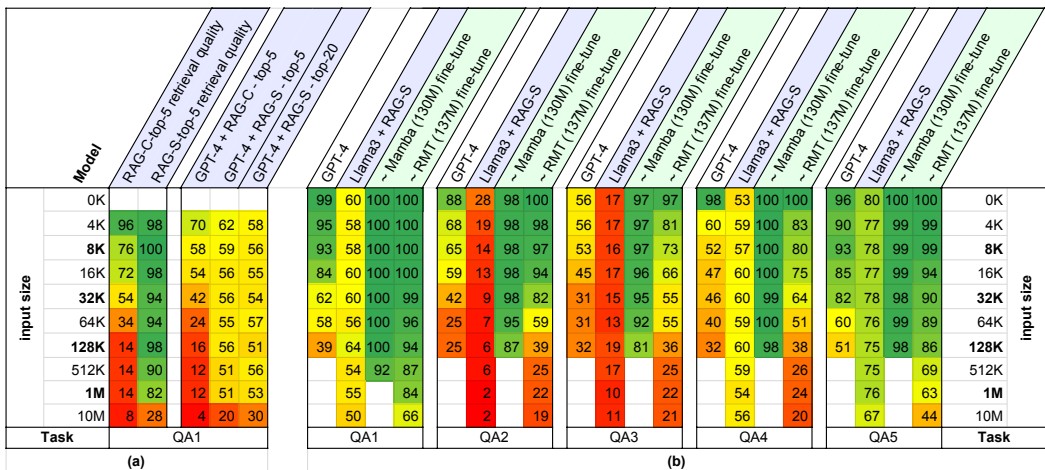

Figure 3: **Fine-tuning but not RAG solves BABILong.** a) RAG on QA1 task. Retrieval by chunks with size 512 tokens (RAG-C) fails to improve GPT-4 and Llama-3 performance on long-context tasks. RAG-S, which retrieves by sentences achieves better results, but further increasing the number of retrieved sentences from top-5 to top-20 does not help. b) Task specific fine-tuning. Finetuned Mamba achieves the best overall results, greatly outperforming RAG models. However, processing sequences longer than 128k is extremely slow due to technical limitations. On the other hand, RMT shines on extremely long sequences, managing to keep high accuracy up to 10M tokens.

results on lengths up to 64K, but unfortunately we were not able to run it on 128K tokens. Phi-3-mini drops significantly from 64K to 128K, reaching less than 10%, while Phi-3-medium maintains 30% at 128K. Jamba-v1 and Phi-3-mini show close results, but Jamba-v1 does not have drop at 128K and shows 34% on this length. Command-R and Phi-3-medium are the most robust to longer contexts, but start to lose performance more sharply at 128K. Phi-3-medium and Command-R show results very close to GPT-4 at 32K+ contexts.

We added results for models released since June 2024, including LLama 3.1, Phi 3.5, Qwen 2.5, and GPT-4o-mini. All claim to support 128K context length. Phi-3.5-mini shows improvements mainly for contexts up to 16K. The new Phi-3.5-MoE performs similarly to Phi-3-medium, but with only 6.6B active parameters compared to 14B in Phi-3-medium. LLama-3.1 models show significant improvement: LLama-3.1-8B matches GPT-4o-mini, while LLama-3.1-70B outperforms GPT-4 on longer contexts. Qwen-2.5 models outperform LLama-3.1 of similar size and achieve the best results of all evaluated open LLMs on BABILong.

Most of the new models use multistage pre-training with progressively increasing sequence lengths. For example, LLama-3.1 (Dubey et al., 2024) is pre-trained in six stages from 8K to 128K, and only proceeds to larger lengths if it maintains high short context scores and successfully solves the needle-in-a-haystack task. During supervised fine-tuning, LLama-3.1 models mix short context data with synthetic long context data, including question-answering, summarization, and code tasks.

## 3.2 Retrieval-Augmented Generation Does Not Perform Well on BABILong

Retrieval-Augmented Generation (RAG) is a popular solution for language models to handle large amounts of text. In RAG relevant parts of texts are retrieved from a large dataset on the first stage. Then, the language model uses input augmented with retrieved texts to generate the final response. In the case of BABILong, we expect RAG to extract all the facts relevant to a question from a long input text and then place them in the context of the model.

We experiment with two options: (1) retrieval by chunks of size 512 tokens, denoted RAG-C and (2) retrieval by sentences, called RAG-S. For details of evaluation and RAG pipelines with GPT4 and Llama-3 please refer to Appendix G. The findings from the QA1 task, depicted in Figure 3a, indicate that retrieval performance using sentence chunks is superior to that of 512-token segments, with a notable decrease in accuracy observed already after 16k token context length. However, this superiority is task-specific and may not translate effectively to real-world applications due to the potential for information loss in smaller chunks.

The RAG pipeline with GPT-4-turbo shows scalable but weak performance on BABILong for sentence embeddings and poor scalability with chunk embeddings (see Fig. 3a). The weak performance of RAG might be attributable to the temporal dependencies inherent in the task, where the relevant fact is positioned at the end of the text. In QA2 and QA3, retrieval fails dramatically with accuracy plummeting below random guessing. This lack of success is attributable to the specific demands of these tasks, which require the retrieval of multiple (two or three) supporting facts to generate accurate responses. For example, in instances where the key facts are "Mary got the milk there." and "Mary travelled to the hallway.", with the query being "Where is the milk?", the retrieval system may successfully identify the first fact but fail to retrieve the second due to insufficient similarity between the question and the latter fact. The default retrieval algorithm's lack of temporal consideration and limitations in the similarity function underscore the necessity for additional methods in tasks with multi-hop reasoning and temporal dynamics.

## 3.3 Fine-Tuning Models on BABILong

We performed fine-tuning experiments with GPT-3.5-Turbo, Mistral-7B-Instruct-v0.2, RMT and ARMT with GPT-2 (137M) backbone, and Mamba (130M) models. Fine-tuning results are in Figure 3b and Appendix I Figure 9.

RMT with a GPT-2 (Radford et al., 2019) backbone model is trained on each task individually with a segment size of 512 and memory size of 16. ARMT with GPT-2 used 10 memory tokens (Rodkin et al., 2024). Train and evaluation splits of each task contain 10000 and 1000 samples, respectively, with a number of facts in each sample ranging from 2 to 320 depending on the task. A curriculum strategy is employed, initiating training on short sequences that fit in a single segment and then gradually introducing longer sequences once the training converges. During each curriculum stage $n$, sequence lengths are randomly sampled from 1 to $n$ segments. We provide details of training and evaluation procedures in Appendix C.

RMT and ARMT models trained on 32 segments totalling in 16K tokens demonstrates strong performance on this length. Notably, recurrent memory models outperform GPT-4 significantly, underscoring the efficiency of memory mechanism in processing long context. Even more importantly, the power of recurrent models extends to sequences longer than the training size. RMT shows consistent performance on longer sequences, up to 128k tokens, with only a marginal quality degradation. Surprisingly, with context sizes scaling to 1 million, 10 million tokens, and even 11.1 million tokens, which is over 600 times of the training length. While ARMT successfully scales even further, reaching up to 50 million tokens (Rodkin et al., 2024). Recurrent memory models persistently outperform the larger counterparts utilizing RAG.

Finetuned recurrent models, Mamba, RMT and ARMT perform equally well on QA1, however due to the technical limitations of the Mamba implementation, the inference beyond 128k was extremely slow, which makes it nearly impossible to process longer sequences. Recurrent Memory models greatly outperform retrieval-augmented models and are able to process sequences up to 10M and 50M tokens much faster than Mamba. However, Mamba has an edge in complex tasks such as remembering a large number of facts in QA3 (Table 4).

We evaluated GPT-3.5 and Mistral-7B models fine-tuned with 1000 samples from QA1 for 3 epochs. The evaluation results are shown in Appendix I Figure 9. Fine-tuning dramatically improves performance for longer contexts making scores uniform across all input sizes. Still, these results are behind of fine-tuned Mamba, RMT and ARMT.

## 3.4 BABILong and Other Benchmarks

Here, we analyze how models performance on BABILong benchmark differs from MMLU (Hendrycks et al., 2020) and RULER (Hsieh et al., 2024). The MMLU benchmark measures various branches of knowledge in LLMs, whereas RULER, a recently proposed long-context benchmark, shares a similar "needle-in-a-haystack" concept with BABILong. One notable difference is that RULER's "needles" (such as adjectives, nouns, numbers, uuids) and long "haystack" contexts are more synthetic, consisting of randomly repeated sentences, except for tasks based on the SQuAD (Rajpurkar et al., 2016) and HotPotQA (Yang et al., 2018) datasets or using Paul Graham essays.

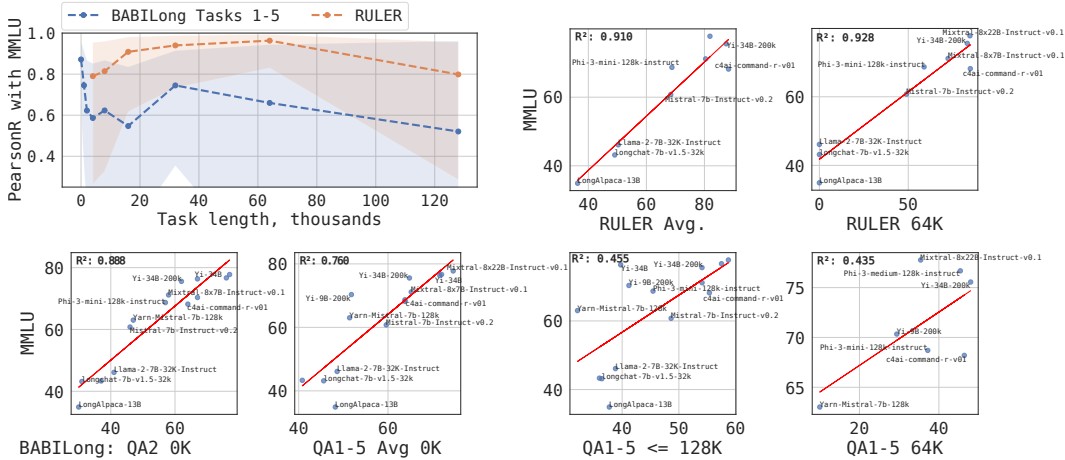

Figure 4: **BABILong is similar to MMLU (Hendrycks et al., 2020) on short lengths and captures differences in models behavior for longer contexts.** MMLU is a relatively short benchmark, with samples up to 1k tokens in length. BABILong has a higher correlation with MMLU on short contexts (0K) than RULER (Hsieh et al., 2024). However, RULER maintains a high correlation regardless of task length, with an even higher correlation at 64K, while BABILong's correlation with MMLU decreases with length. This may indicate that BABILong is better at capturing differences in models behavior at different context lengths.

We collect results from multiple models on MMLU, BABILong, and RULER at lengths ranging from 0K (BABILong without texts from PG19) to 128K tokens. In the upper-left part of Figure 4, we show the correlation between scores on BABILong and RULER for different task lengths with those on the MMLU benchmark. At shorter lengths, BABILong exhibits a high correlation with MMLU, which diminishes as the length increases. Conversely, RULER shows a nearly constant correlation with MMLU, regardless of the length. The best correlated RULER lengths with MMLU are 64K and the average of all lengths (<=128K). In contrast, the highest correlation of BABILong scores with MMLU is at length 0K, which is expected since MMLU is a relatively short benchmark with examples up to 1K tokens. Comparing the correlations of BABILong with MMLU at the most correlated RULER lengths (<=128K and 64K) shows much lower values: 0.928 vs. 0.435 and 0.910 vs. 0.455, respectively.

These results show that BABILong can detect differences in models behavior starting from lengths as small as 2K tokens, while RULER requires lengths of at least 128K tokens to show significant differentiation from relatively short MMLU benchmark.

## 4 Related Work on Long Context Benchmarks and Datasets

Long Range Arena (LRA) (Tay et al., 2021) was a one of the pioneering benchmarks for long context modeling. LRA is a set of tasks with lengths from 1 to 16 thousand tokens. However, it mainly consists of very specific tasks such as ListOps (2k tokens), Byte-Level Text Classification (4k tokens) and Byte-Level Text Retrieval (8k tokens), and others that are less related to NLP. They are not well suited for evaluating of modern LLMs without fine-tuning on these tasks and cannot fully represent the capabilites of LLMs that can handle 100k+ tokens.

A new set of datasets and benchmarks specifically designed to test the ability of LLMs to handle long contexts has been proposed. The LongBench dataset (Bai et al., 2023) contains 6 types of real and synthetic problems, ranging from summarization and multidoc QA to code completion. The average sample lengths in LongBench are 6k and 13k tokens for English and Chinese respectively, with 40k tokens at max. Scrolls and ZeroSCROLLS (Shaham et al., 2022, 2023) consist of QA, classification, summarization tasks and have higher average lengths ranging from 1.7k to 49.3k tokens. L-Eval (An et al., 2023) mostly combines 20 smaller long sequence datasets and adds 4 newly annotated tasks, with query-response pairs encompassing diverse question styles and domains. The average length of examples for L-Eval varies from 3 to 60 thousand tokens. Some of the benchmarks are focusing on evaluation of in-context learning and instruction following, such as LongAlign and LongBench-chat (Bai et al., 2024), ZeroScrolls, LongICLBench (Li et al., 2024).

There are other long-context datasets that primarily consist of QA and summarization tasks over texts from Wiki, arXiv, novels, movie and TV scripts, or other sources, e.g., InfinityBench (Zhang et al., 2024b), Loogle (Li et al., 2023b), Bamboo (Dong et al., 2023), LVEval (Yuan et al., 2024), NovelQA (Wang et al., 2024b), Marathon (Zhang et al., 2023), XL$^2$-Bench (Ni et al., 2024), DocFinQA (Reddy et al., 2024), or ChapterBreak (Sun et al., 2022), Ada-LEval (Wang et al., 2024a) that evaluate operations with text chunks, or move to multimodal tasks in MileBench (Song et al., 2024a). These datasets vary in length, with maximum sample lengths of 636K tokens in ChapterBreak and average lengths reaching 200K tokens in InfinityBench, NovelQA, LVEval, and some subsets of XL$^2$-Bench.

Further extending of benchmarks' length with real and human annotated data is very challenging. Therefore, "needle-in-a-haystack" inspired benchmarks were proposed. Following LLMTest [6] with magic numbers as needles in Paul Graham essays as haystack, passkey and key-value retrieval tasks are part of InfinityBench (Zhang et al., 2024b). Counting-Stars (Song et al., 2024b) suggests to insert multiple sentences about little penguins that count stars into the same essays for English or The Story of the Stone for the Chinese language. The task is to answer questions based on these "needle" sentences. RULER (Hsieh et al., 2024) extends "needle-in-a-haystack" with multiple types and amount of "needles". RULER and Counting-Stars introduce new task categories such as multi-hop tracing and aggregation to test models beyond searching from context.

Some benchmarks have pre-defined length bins, such as LongBench (0-4k, 4k-8k, 8k+), Ada-LEval (2k-128k), LVEval (16k, 32k, 64k, 128k, 256k), Bamboo (4k, 16k), S3Eval (2k, 4k, 8k, 16k) (Lei et al., 2023). A number of benchmarks, including RULER, CountingStars, Ada-LEval, and S3Eval, can be generated at required lengths. All mentioned datasets are mostly in English with some of them covering Chinese language (LongBench, InfinityBench, LongAlign, Counting Stars, CLongEval (Qiu et al., 2024), LV-Eval, XL$^2$-Bench).

BABILong focuses on natural language reasoning over multiple facts distributed in very large textual corpora. Compared to existing approaches it provides more tasks and more natural and deceptive mixing of information into background documents. BABILong consists of diverse set of 20 tasks that cover different capabilities including multi-hop tracing, aggregation over needles and extending them with basic deduction and induction, time, positional, and size reasoning, and path finding. The benchmark goes with predefined splits up to unprecedented 10M token length. Lengths beyond 10M tokens could be generated and we test models up to 50M tokens. Furthermore, while we evaluate models on English-only tasks from bAbI (Weston et al., 2016) adding new languages is straightforward.

## Conclusions

In this work, we introduced the BABILong, a diverse and scalable benchmark designed to bridge the gap in evaluating large language models (LLMs) across extensive context lengths. Our experiments demonstrate that BABILong offers a more representative evaluation framework for long-context reasoning among the existing benchmarks. The analysis of correlation with other benchmarks further validates BABILong's ability to pose a significant challenge for large language models to maintain performance as context lengths scale up. The BABILong benchmark offers algorithmically adaptable document length and facts placement, includes predefined sets of bins ranging from 0k to 10M tokens. Facts in BABILong could be generated making it leak-proof for future LLMs. It consists of a set of 20 diverse tasks covering reasoning tasks, including fact chaining, simple induction, deduction, counting, and handling lists/sets. Compared to other benchmarks, BABILong shows high correlation on short contexts with MMLU and diverges from it as lengths increases.

Our findings reveal limitations of popular open-source LLMs as well as GPT-4, Gemini 1.5 Pro and RAG solutions regarding effective long context utilization. Their performance heavily relies on the first 5-25% of the input, highlighting the need for improved context processing mechanisms. The recent open-source models LLama-3.1 and Qwen-2.5 show the best performance of pre-trained LLMs, with LLama-3.1 using pre-training and supervised fine-tuning on long context data up to 128K tokens. BABILong fine-tuning experiments show that tasks from BABILong are solvable even by relatively small models like RMT & ARMT with GPT-2 (137M) and Mamba (130M). Fine-tuning improves the performance of GPT-3.5-Turbo and Mistral-7B, but their context lengths remain limited

---

[6] https://github.com/gkamradt/LLMTest_NeedleInAHaystack

to 16K and 32K, respectively. Among the evaluated models, Mamba and recurrent transformers achieve the strongest results. However, Mamba is hard to infer on lengths more than 128K tokens, while RMT and ARMT enables the processing of lengths up to 50 million tokens.

## Limitations

The BABILong benchmark uses background texts to hide facts in them. In our experiments, we only tried PG19 and Wiki as background text sources. Other background texts may have a different effect on the results. PG19 and Wiki were chosen because they contain natural narratives and facts about people, in a way similar to bAbI tasks. Interference between similar facts in the background text can make the benchmark even more difficult.

In GPT-4 and LLama-3 with RAG experiments, we do not optimize the retrieval component. We tried several prompts experiments with LLMs, but the ones that we selected could be suboptimal. We provide them in Appendix J and in our GitHub repository.

The current version of the dataset reuses parameters for fact generation from the bAbI (Weston et al., 2016). As the initial work used vocabularies of limited size, this results in a low variety of names and objects within the facts. This limitation makes the BABILong tasks easier for fine-tuned models, as they can quickly learn specific tokens that differentiate facts from background text. This issue is partially mitigated by generating distractor facts using the same vocabularies. Enhancing the dataset's vocabulary in future versions could easily address this limitation.

We could also use other sources of facts and questions (e.g., reasoning datasets such as Rule-taker (Clark et al., 2020), ProofWriter (Tafjord et al., 2021), FOLIO (Han et al., 2022), PrOn-toQA (Saparov & He, 2023), LogicNLI (Tian et al., 2021), and DELTA (Poulis et al., 2024)), mixing samples of question-answering datasets with background text from the same domain, or using LLMs to generate questions about statements that belong to the original documents. Keeping the same framework as in BABILong, this will lead to more complex and real-world scenarios.

Although recurrent approaches, like RMT, are hindered by their sequential nature, resulting in reduced parallelizability, they compensate by constant memory requirements, but it is also their limitation as storage capacity of the model is finite. However, BABILong is solvable by this type of models.

## Author contributions

Y.K., A.B. and M.B. conceived of the idea. A.B. prepared the dataset. Y.K. and A.B. scored open-source LLMs, A.B. scored RMT, I.R. scored ARMT, Mamba and Gemini pro 1.5, P.A. scored Mistal models and RAG pipelines, D.S. scored GPT-4 and GPT-3.5, A.S. fine-tuned and scored GPT-3.5, A.B. finetuned and scored Mistral. D.S. created public leaderboard. A.B. populated and maintain leaderboard. Y.K. performed study to compare MMLU, RULER and BABILong. Y.K., A.B. and M.B. aggregated and analyzed scoring results. All authors contributed to discussions of results. Y.K., A.B. and P.A. wrote the first draft of the manuscript. All the authors contributed in extending manuscript with their results. M.B. took the lead in structuring and editing manuscript towards the final version.

## Acknowledgments and Disclosure of Funding

We are thankful to SberDevices for granting us access to additional computational resources. This work was partially supported by a grant for research centers, provided by the Analytical Center for the Government of the Russian Federation in accordance with the subsidy agreement (agreement identifier 000000D730324P540002) and the agreement with the Moscow Institute of Physics and Technology dated November 1, 2021 No. 70-2021-00138.

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

| Appendix | Content |
|----------|---------|
| Appendix A | Code and Data Availability |
| Appendix B | Related Work on Long Context Models |
| Appendix C | Details on RMT and Mamba fine-tuning on BABILong |
| Appendix D | Detailed LLMs evaluation on BABILong QA1-5 tasks |
| Appendix E | Gemini Evaluation |
| Appendix F | BABILong Dataset Statistics |
| Appendix G | Details of the RAG Pipeline |
| Appendix H | Recurrent Memory Transformer Analysis |
| Appendix I | LLMs fine-tuning results |
| Appendix J | Prompts Used to Benchmark Large Language Models |
| Appendix K | Analysis of LLM Performance for Different Locations of the Supporting Facts |
| Appendix L | BABILong Task Examples |
| Appendix M | Author Statement |
| Appendix N | BABILong Datasheet |

## A  Code and Data Availability

Code for generating data and evaluating models is available at `https://github.com/booydar/babilong`.

We also provide pre-generated evaluation data hosted on HuggingFace datasets. The evaluation sets include 100 samples per length and per task, with lengths from 0k (no background text from PG-19) to 10 million tokens: `https://huggingface.co/datasets/RMT-team/babilong` and 1000 samples per length and per task with lengths from 0k to 128k tokens: `https://huggingface.co/datasets/RMT-team/RMT-team/babilong-1k-samples`.

The croissant metadata for both evaluation sets is provided by HuggingFace:

`https://huggingface.co/api/datasets/RMT-team/babilong/croissant`

`https://huggingface.co/api/datasets/RMT-team/babilong-1k-samples/croissant`.

Our code is released under the Apache 2.0 License. We use data from the PG-19 corpora (Rae et al., 2020) (Apache 2.0 License[7]) and the bAbI dataset (Weston et al., 2016) (BSD License[8]).

### A.1  Reproducibility

Our code includes data generation, metrics, and the evaluation pipeline used to benchmark models. Additionally, we release the predictions of all models used in our study to ensure that all reported results can be reproduced and verified: `https://github.com/booydar/babilong/tree/predictions_06_2024`.

## B  Related Work on Long Context Models

**Approaches to long context processing**  In retrieval augmented generation (RAG), a language model is combined with a separate module, called a retriever. Given a specific request, the retriever finds a set of relevant parts from a dedicated data storage. Then parts selected by the retriever along with the input are incorporated by the language model to make predictions. Many different implementations of the retrieval mechanism have been proposed (Guu et al., 2020; Borgeaud et al., 2022; Shi et al., 2023). Some works focus on directly retrieving predictions (Khandelwal et al.,

---

[7]`https://github.com/google-deepmind/pg19`
[8]`https://github.com/facebookarchive/bAbI-tasks/blob/master/LICENSE.md`

2019). Other works retrieve individual input tokens or text segments and add them to the LM input (Guu et al., 2020; Borgeaud et al., 2022). For example, in REALM (Guu et al., 2020) whole text segments are retrieved and appended to the input to improve masked language modeling. In Memorizing Transformer (Wu et al., 2022b), the retriever returns cached (key, value) pairs saved from previous training steps of the language model. In Retrieval-Pretrained Transformer (Rubin & Berant, 2023), an LM component processes long documents in chunks and a retriever finds relevant chunks. Representations of retrieved chunks are fused with current chunk in the LM component, and both the LM and retrieval parts are trained jointly. AutoCompressor (Chevalier et al., 2023) combines RMT-like (Bulatov et al., 2022) approach with retrieval from external corpora. AutoCompressor is first used to produce memory tokens (or summary vectors) for text chunks. Next, off-the-shelf retriever is used and corresponding chunk's memory tokens are added to the context of the model.

In this work, we augment the Recurrent Memory Transformer (Bulatov et al., 2024) with the ability to retrieve its own past memory tokens. As far as we know, this is the first combination of a recurrent transformer with a trainable retrieval mechanism.

Recurrence is another mechanism to deal with long context (Graves et al., 2014; Voelker et al., 2019; Sorokin et al., 2022). Instead of processing the entire context, a recurrent model breaks it down into smaller segments. The recurrent hidden state acts as an aggregator of information from past segments of the sequence. Attending to a memory state is much cheaper than to all contexts. Many different architectures adding recurrence to transformers have been proposed (Wu et al., 2022a; Lei et al., 2020; Fan et al., 2020). For example, Compressive Transformer (Rae et al., 2020) updates recurrent memory by compressing hidden activation's from the previous segment to a smaller set of representations. Recurrent Memory Transformer (Bulatov et al., 2022) recurrently passes states of special memory tokens added to the input of Transformer.

Activation Beacon (Zhang et al., 2024a) compresses activations from prior segments using separate parameters and integrates a sliding window mechanism, handling up to 400k tokens. Temporal Latent Bottleneck (Didolkar et al., 2022) Transformer splits computation into two streams: recurrent slow stream and fast stream with self-attention between tokens. Block-Recurrent Transformer (Hutchins et al., 2022) employs LSTM-style (Hochreiter & Schmidhuber, 1997) gates to update its recurrent state.

We use RMT in our experiments because of its simplicity, plug-and-play compatibility with pre-trained transformer-based language models, and promising scaling capabilities (Bulatov et al., 2024).

Big Bird (Zaheer et al., 2020), Longformer (Beltagy et al., 2020), LongNet (Ding et al., 2023) help extend context length for Transformers by switching from full self-attention to sparse self-attention mechanisms with linear complexity. Works like RWKV (Peng et al., 2023a), S4 (Gu et al., 2021), Mamba (Gu & Dao, 2023), take another approach and focus on advancing recurrent networks to reach high parallelism levels available to Transformers while retaining the linear complexity of RNN. These works show promising results on long sequences but are still lagging behind the best transformer models in natural language processing tasks. Mamba, however, seeks to bridge this gap.

## C  Details on RMT, ARMT, and Mamba fine-tuning and evaluation on BABILong

We used the GPT-2 (Radford et al., 2019) (137M) model (`https://huggingface.co/GPT-2`) as the backbone for RMT and ARMT. The segment size was fixed at 512 tokens, and the model was augmented with 16 memory tokens for RMT and 10 memory tokens in ARMT. Finetuning was conducted on BABILong using a curriculum schedule with progressively increasing sequence lengths: 1, 2, 4, 6, 8, 16 and 32 segments. ARMT used 2-3-5-8-16-32. For each curriculum step $n$ we chose the number of segments randomly from 1 to $n$ for every batch to prevent overfitting to a certain context size. We maintained a fixed batch size of 64 and the AdamW (Loshchilov & Hutter, 2019) optimizer with learning rate in range {5e-05, 3e-05}, a linear schedule and 1000 warmup steps. In ARMT experiments, the learning rate was set to 1e-04. We also consider the memory-dimension parameter for ARMT to be 64 and we use non-linearity DPFP-3 (Schlag et al., 2021). Each curriculum stage had a maximum of 10,000 steps with early stopping if metrics stop increasing. The weight decay value was set to 0.01, and no gradient stopping was used. Training was performed on 1-4 Nvidia A100 or H100 GPUs with the duration of each curriculum stage ranging from 40 minutes to 20 hours.

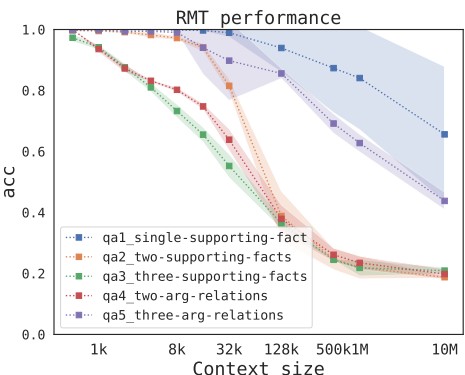

Figure 5: RMT performance on five BABILong tasks varies between training seeds. The plot represents average performance and standard deviation across three training runs.

For each experiment we conducted three runs with different memory intitalizations and dataset shuffles. As shown in Figure 5, performance on context lengths, exceeding ones seen during training, may vary across different runs. This suggests that the early stopping criterion based on short-context accuracy may not be optimal. To reduce deviations between runs and enhance overall performance, techniques such as improving early stopping, gradient truncation and training on longer sequences can be employed.

To fine-tune mamba-130m, we used the exact same curriculum approach, with a randomly selected number of segments that gradually increased. Throughout every curriculum step, the batch size remained constant at 128. We employed the AdamW optimizer with a linear schedule, weight decay of 2.0, gradient clipping of 1.0, learning rate of 3e–4, and a warmup step count of 10% of the total training steps. The model was trained for 10,000 steps in each curriculum stage except for the last one, which had 32 segments, where it was trained for 15,000 steps. 4 NVidia H100 GPUs were used for the training, and the overall training process for every task from BABILong took 2 to 3 days to complete.

To evaluate recurrent models on the BABILong benchmark we use the full test set for sequences up to 1M tokens, for 10M tokens we only provide results on averaged over 100 samples. As shown in Table 3, evaluation time grows linearly with context length. We fix a random seed used to sample background texts from PG19 for the test set. However, the seed was not fixed to avoid overfitting to specific sampled texts during training.

Table 3: Time required for processing 1000 BABILong samples with RMT using a single A100 80Gb GPU, including input data processing.

| CONTEXT SIZE | 4K | 32K | 128K | 1M |
|---|---|---|---|---|
| PROCESSING TIME, MINUTES | 4 | 30 | 80 | 315 |

## D  Detailed LLMs evaluation on BABILong QA1-5 tasks

Here we present the complete results of LLMs evaluation. Table 4 showcases the performance of 38 models across the first five tasks. Comparing the tasks in the table makes evident the difference in task complexity for language models. QA1 and QA5 are the easiest, with most models achieving over 70% accuracy for the 0k split. QA4 is significantly more challenging, and only 5 models can reach this level of performance. QA2 and QA3 pose even greater challenges for most models.

The number of parameters positively impacts accuracy on the shortest 0k split. Among not-finetuned models, GPT-4, Phi-3-medium, Qwen, Jamba, Command-R, Yi-34B and Mixtral 8x22B consistently outperform smaller models. Notably, RWKV and Mamba-2.8B also demonstrate strong performance on QA2 and QA3. However, as the context length increases, some of the largest models lose their advantage over smaller ones.

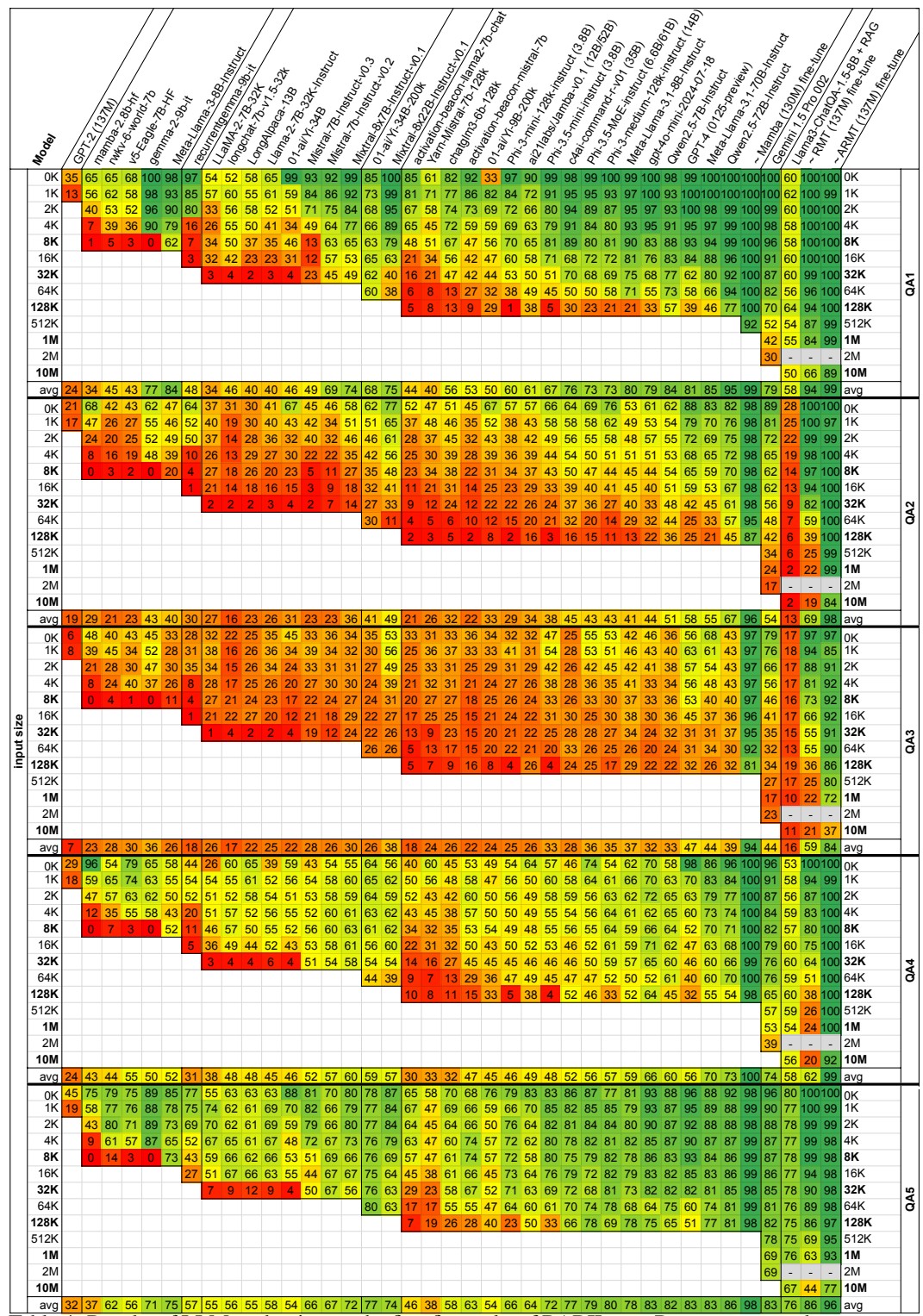

Table 4: Results of LLM evaluation on the first five tasks of BABILong. Rows correspond to sequence lengths, columns denote models, and each section represents a separate task from QA1 to QA5. Each number indicates the average accuracy of the model at a given sequence length, calculated over 1000 samples for lengths up to 32k tokens, and over 100 samples for longer lengths.

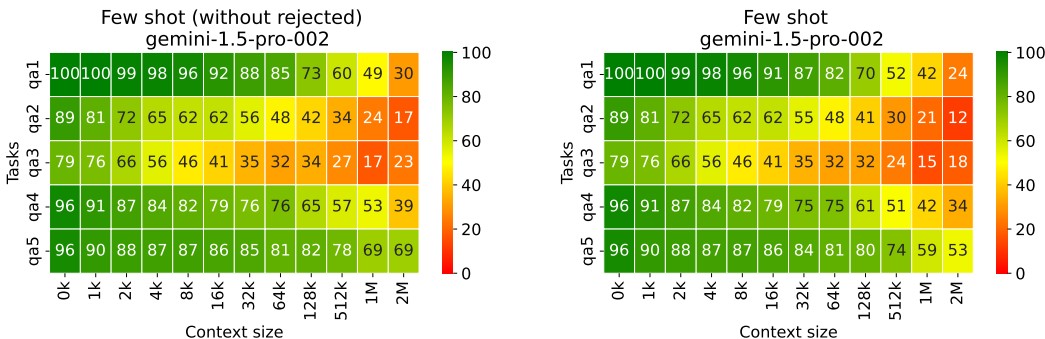

Figure 6: Results of Gemini 1.5 Pro 002 evaluations. Built-in content safety filtering causes up to 14% of requests to be denied with increased context size. Without rejected means that we removed from evaluation all cases where the model refused to answer. Few-shot means that we used instructions with in-context examples.

Retreival-augmented Llama-3 has a strong advantage of being able to perform on any context length up to 10M tokens. On QA4 and QA5 retrieval allows to match and even surpass weaker competitors on longer context sizes. However, on QA2 and QA3 this approach fails dramatically. The reason for this performance drop lies in inability of retrieval to maintain the order of found sentences, complicating the task for the underlying Llama. Additionally, relevant sentences in these tasks are not always semantically similar to the question, preventing the model from retrieving all necessary facts for correct reasoning.

It is important to note, that all BABILong tasks are in practice solvable even with smaller models. Finetuned RMT, ARMT, and Mamba achieve outstanding scores across most sequence lengths, significantly outperforming LLMs despite having up to 100 times ewer parameters. Mamba has an advantage on medium-length sequences, but recurrent memory models (RMT and ARMT) excel in processing much larger sequences up to 10M tokens.

## E   Gemini Evaluation

We evaluated the Gemini 1.5 Pro 002 model on the QA1 task of BABILong. We were requesting model responses via API, some of the requests were denied due to built-in content safety filtering, even with BLOCK_NONE set. In Fig. 1b we only report results for requests where the model did not refused to response. We present the full results in Figure 6. We found that as the context size increases, the model can refuse to respond up to 14% of the time. We used 1000 samples for lengths up to 32K, for larger lengths we used 100 samples per length.

## F   BABILong Dataset Statistics

The proposed benchmark includes 20 diverse tasks, ranging from simple "needle in a haystack" scenarios with distractor facts to more complex tasks that require counting, logical reasoning, or spatial reasoning. The Figure 7 evaluates the complexity of the base short versions of these tasks. Tasks such as QA1, QA5, and QA10 are generally easier for most models, whereas QA7, QA15, and QA19 are the most challenging. The plot clearly shows that the number of facts needed for reasoning significantly impacts task complexity, as performance gradually declines from QA1 to QA2 and QA3, which differ in the number of supporting facts. The distribution of task labels is shown in Table 6.

BABILong is a generative benchmark, designed to be scalable with increasing length of language models. The same bAbI task can be scaled to any desired length in tokens by adding a sufficient number of distractor sentences. For reproducibility, we pre-generate dataset splits for several fixed lengths: 0k (tasks with no distractor sentences), 4k, 8k, 16k, 32k, 64k, 128k, 512k, 1M and 10M tokens. The length in tokens is measured using the classic GPT-2 tokenizer, which is close in fertility to the popular GPT-4 tokenizer. As shown in Table 5, the number of tokens for tokenizers of different

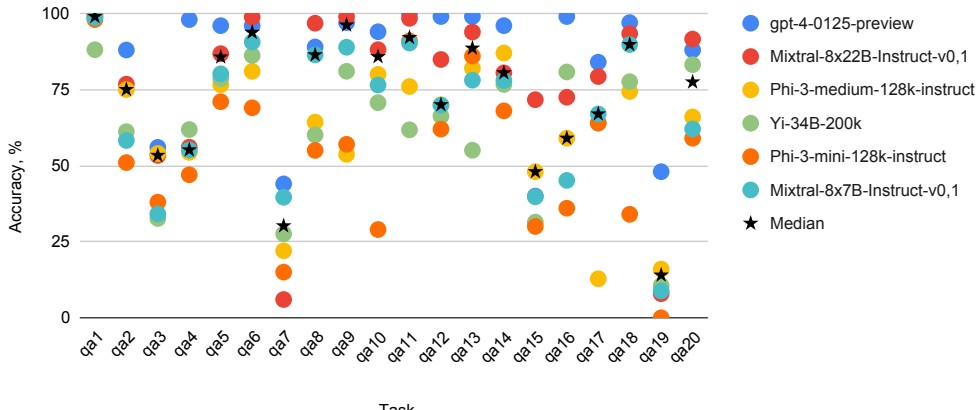

Figure 7: The performance of LLMs on the bAbI (BABILong without distractor text) depends significantly on the task complexity. Each dot represents the average accuracy of the model on one thousand samples of the given task. The median accuracy across all models is denoted by black stars.

models may differ for samples in the same split. However considering the trade-off between the sequence length and embedding layer size, we believe the comparison remains fair.

Table 5: Token count for various models across selected tasks. We measure the length of BABILong samples using the conservative GPT-2 tokenizer. Actual token sizes may vary depending on the model tokenizer.

| MODEL | 0K | 4K | 16K | 64K | 128K |
|---|---|---|---|---|---|
| GPT-4 | 120 | 3544 | 15071 | 61343 | 123367 |
| GPT-2 | 120 | 3700 | 15699 | 63698 | 127695 |
| LLAMA-2 | 135 | 3942 | 16757 | 68110 | 137222 |
| MISTRAL | 128 | 3863 | 16438 | 66862 | 134592 |
| WORDS | 98 | 2548 | 10789 | 44180 | 88592 |
| SYMBOLS | 561 | 14507 | 61452 | 251947 | 507598 |

Table 6: The distribution of labels in first five BABILong tasks, % of all samples.

| | LABEL1 | LABEL2 | LABEL3 | LABEL4 | LABEL5 | LABEL6 | LABEL7 |
|---|---|---|---|---|---|---|---|
| QA1 | 15.4 | 14.9 | 15.7 | 18.7 | 18.2 | 17.1 | |
| QA2 | 15.9 | 18.7 | 16.7 | 16.5 | 17.5 | 14.6 | |
| QA3 | 13.3 | 18.4 | 21.5 | 14.6 | 15.4 | 16.7 | |
| QA4 | 15.6 | 17.7 | 16.6 | 17.1 | 15.3 | 17.6 | |
| QA5 | 9.5 | 18.8 | 12.9 | 16.4 | 13.6 | 18.9 | 9.8 |

# G   Details of the RAG Pipeline

For the GPT4-RAG pipelines, we employed the FAISS (Douze et al., 2024) vector database, using Langchain (Chase, 2022), for our experimental RAG setup. We utilized the 'text-embedding-ada-002' model for generating text embeddings. Our methodology encompassed two distinct approaches for text chunking: firstly, segmentation by sentences utilizing the NLTK library, and secondly, division into segments of 512 tokens each. We adopted a binary metric for evaluating retrieval accuracy, where the criterion was the presence or absence of relevant facts (singular or multiple, based on the specific task) within the retrieved text chunks. This retrieval accuracy was quantified for the top 5 chunks. Additionally, we assessed the performance of GPT-4-turbo in conjunction with the retrieved facts, specifically focusing on the 'QA1' task. Our experimental scope spanned various context lengths, including 8k, 64k, and 128k tokens for tasks 'QA1' through 'QA5' of the BABILong dataset, with added 4k, 16k, 32k, 500k, 1M and 10M token length for an in-depth analysis of the 'QA1' task. Additionally, we assessed the performance of RAG on the 'QA1' task, utilizing

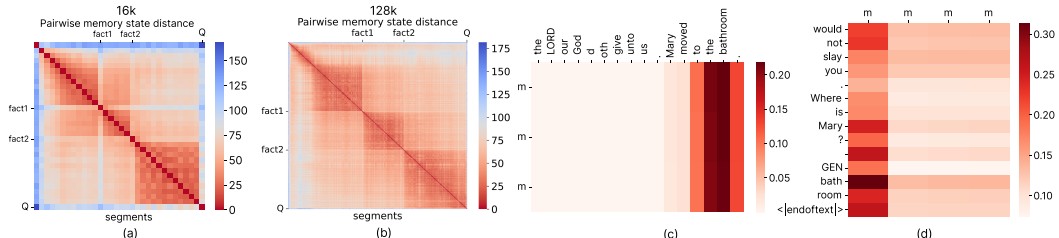

Figure 8: **RMT learns to detect and store relevant facts using memory.** Heatmaps (a) and (b) represent pairwise distances between memory states on QA1 with context size 16k (a) and 128k (b). Distant states are marked with blue color and similar ones with red. Changes in memory mainly occurs when the model meets a new fact, which indicates model adaptation to distinguishing and storing facts in memory. Memory attention maps (c) and (d) when RMT writes the fact to memory (c) and then reads it when answering the question (d). The intensity of red color corresponds to the amount of attention between the query on the left and key on the top.

precomputed Wikipedia embeddings[9] instead of pg-19 with an average embedding size of 250 tokens. This evaluation aimed to determine the influence of embedding size and noise characteristics on model performance. For each task, we maintained a consistent sample size of 50 across different context lengths. For the Llama3 + RAG pipeline we used the 'nvidia/Llama3-ChatQA-1.5-8B' as the language model (Liu et al., 2024b) and the 'nvidia/dragon-multiturn-query-encoder' for context embedding. Another difference is that we did not use any caching and Wikipedia embeddings unlike with GPT-4.

# H    Recurrent Memory Transformer Analysis

To understand how recurrent models consistently retain their performance over extremely long sequences, we analyze the RMT memory states and attention patterns on the QA1 task. We evaluate RMT trained on 32 segments or approximately 16k tokens on a single sample with two facts, see Figure 8 (a) and (b). For both sequence lengths 16k and 128k the memory states exhibit a consistent pattern. In the absence of fact in input, the memory remains similar to its initial states, but the introduction of fact leads to visible change in the memory state. This indicates that the model learned to distinguish important facts from the background text and preserving them in memory until a question appears. The operations with memory are represented by distinctive patterns on attention maps, specifically, the process of writing a new fact to memory Figure 8 (c) and reading from memory to answer a question (d). This visual demonstration supports the intuition of learning distinct memory operations when dealing with information scattered across extended contextual spans.

# I    LLMs fine-tuning results

Results for GPT-3.5 and Mistral-7B fine-tuning are shown on the Fig. 9.

# J    Prompts Used to Benchmark Large Language Models

Each prompt starts with the description of the task followed by several examples inside the <example> </example> tags. The next section inside <context> </context> tags contains an instance of the task. We additionally duplicate the question with the QUESTION mark, in order for the model recognize the question in the large input prompts. The last sentences specify the required response format.

---

[9]https://huggingface.co/datasets/Supabase/wikipedia-en-embeddings

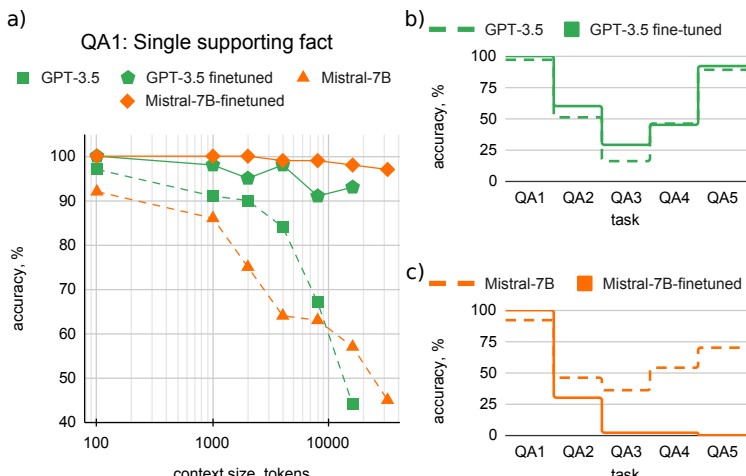

Figure 9: **LLM fine-tuning makes full context effective. a)** After fine-tuning both GPT-3.5 and Mistral-7B significantly improved their scores along context lengths achieving 90% + accuracy on QA1 task. **b)** GPT-3.5 fine-tuned for QA1 task shows improved performance on QA2-QA5 tasks. **c)** Full fine-tuning of smaller Mistral-7B on QA1 results in degraded scores for other tasks (QA2-QA5). No distractor text for b) and c).

---

**QA1 task**

```
I will give you context with the facts about positions of
different persons hidden in some random text and a question.
You need to answer the question based only on the information
from the facts. If a person was in different locations, use the
latest location to answer the question.

<example>
Charlie went to the hallway. Judith come back to the kitchen.
Charlie travelled to balcony. Where is Charlie?
Answer: The most recent location of Charlie is balcony.
</example>

<example>
Alan moved to the garage. Charlie went to the beach. Alan went
to the shop. Rouse travelled to balcony. Where is Alan?
Answer: The most recent location of Alan is shop.
</example>

<context>
{QA1 query with noise}
</context>

QUESTION: {QA1 question}
```

**QA2 task**

```
I give you context with the facts about locations and actions
of different persons hidden in some random text and a question.
You need to answer the question based only on the information
from the facts.

If a person got an item in the first location and travelled to
the second location the item is also in the second location.
If a person dropped an item in the first location and moved to
the second location the item remains in the first location.

<example>
Charlie went to the kitchen. Charlie got a bottle. Charlie
moved to the balcony. Where is the bottle?
Answer: The bottle is in the balcony.
</example>

<example>
Alan moved to the garage. Alan got a screw driver. Alan moved
to the kitchen. Where is the screw driver?
Answer: The screw driver is in the kitchen.
</example>

<context>
{QA2 query with noise}
</context>

QUESTION: {QA2 question}
```

**QA3 task**

```
I give you context with the facts about locations and actions
of different persons hidden in some random text and a question.
You need to answer the question based only on the information
from the facts.

If a person got an item in the first location and travelled to
the second location the item is also in the second location.
If a person dropped an item in the first location and moved to
the second location the item remains in the first location

<example>
John journeyed to the bedroom.Mary grabbed the apple. Mary went
back to the bathroom. Daniel journeyed to the bedroom. Daniel
moved to the garden. Mary travelled to the kitchen. Where was
the apple before the kitchen?
Answer: Before the kitchen the apple was in the bathroom.
</example>

<example>
John went back to the bedroom. John went back to the garden.
John went back to the kitchen. Sandra took the football. Sandra
travelled to the garden. Sandra journeyed to the bedroom. Where
was the football before the bedroom?
Answer: Before the kitchen the football was in the garden.
</example>

<context>
{QA3 query with noise}
</context>

QUESTION: {QA3 question}
```

## QA4 task

```
I will give you context with the facts about different people,
their location and actions, hidden in some random text and a
question.
You need to answer the question based only on the information
from the facts.

<example>
The hallway is south of the kitchen. The bedroom is north of
the kitchen. What is the kitchen south of?
Answer: bedroom
</example>

<example>
The garden is west of the bedroom. The bedroom is west of the
kitchen. What is west of the bedroom?
Answer: garden
</example>

<context>
{QA4 query with noise}
</context>

QUESTION: {QA4 question}
```

## QA5 task

```
I will give you context with the facts about locations and
their relations hidden in some random text and a question. You
need to answer the question based only on the information from
the facts.

<example>
Mary picked up the apple there. Mary gave the apple to Fred.
Mary moved to the bedroom. Bill took the milk there. Who did
Mary give the apple to?
Answer: Fred
</example>

<example>
1 Jeff took the football there. Jeff passed the football to
Fred. Jeff got the milk there. Bill travelled to the bedroom.
Who gave the football?
Answer: Jeff
</example>

<example>
Fred picked up the apple there. Fred handed the apple to Bill.
Bill journeyed to the bedroom. Jeff went back to the garden.
What did Fred give to Bill?
Answer: apple
</example>

<context>
{QA5 query with noise}
</context>

QUESTION: {QA5 question}
```

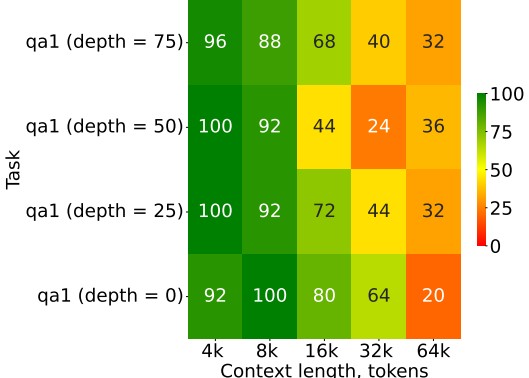

Figure 10: Evaluation results for GPT-4-Turbo with different locations of the facts in the QA1 task.

# K  Analysis of LLM Performance for Different Locations of the Supporting Facts

Fig. 10 shows the evaluation result of the GPT-4-Turbo model when all the facts in task are located in the same quarter of the input query. It is seen that the performance of the model is not the same for different locations of the supporting facts. The most difficult location to identify the facts is in the middle of context which corresponds to the depts = 50 in the Fig. 10.

# L BABILong Task Examples

This section contains samples of the final BABILong dataset for first five tasks. First part of each example displays facts needed to solve the task, second part shows the example with background text with total length up to 512 tokens, and the final part contains question and the desired answer. The tasks differ by the number of facts and the task complexity, testing the ability for multiple reasoning aspects.

**QA1 single-supporting-fact**

**Facts:** Sandra moved to the kitchen. Sandra went back to the garden. Sandra journeyed to the office. Mary moved to the office. Sandra journeyed to the bathroom. Daniel moved to the office. Daniel went back to the kitchen. Mary moved to the hallway.

**Input:** Now this loss of the sense of proportion in human affairs, Sir, is a very bad sign, and a well-nigh infallible indicator of nerve-strain and general overpressure. **Sandra moved to the kitchen.** But I find a yet more unmistakable evidence in support of my contention in the extraordinary emotional sensibility revealed by these headlines whenever some unfortunate person has been sentenced to death for the most commonplace murder. There is clearly a profound conviction that the jury who heard the evidence, the judge who pronounced their verdict of guilty, the only possible conclusion they could reasonable come to, and the HOME SECRETARY who found himself unable to recommend a reprieve, were, one and all, engaged in a cold-blooded conspiracy against a perfectly innocent man. The convict has said to himself, and that seems to be considered sufficient. And so, night after night, the authors of these headlines harrow themselves by announcing such items as "Blank protests his innocence to his Solicitor." "Distressing Scene on the Scaffold." **Sandra went back to the garden.** Consider the strain of all these alterations of hope and despair, repeated time after time, and almost invariably without even the consolation of deferring the fate of their protege by a single hour! **Sandra journeyed to the office.** Is it not too much for the strongest constitution to endure? a service which the society has no right to demand from any of its members? Yes, Sir, whether these devoted servants of the public know it or not, they are running a most frightful risk; the word which hangs above their heads may fall at any moment. **Mary moved to the office. Sandra journeyed to the bathroom. Daniel moved to the office.** Suppose, for example–and it is surely not wholly an imaginary danger I foresee–suppose that some day some event should happen somewhere of real and serious importance. **Daniel went back to the kitchen. Mary moved to the hallway.** Have they left themselves any epithet in reserve capable of expressing their sensations at all adequately? They have not; they have squandered participles and adjectives in such reckless profusion that they will discover they are reduced to the condition of inarticulate bankrupts; and, speaking as a medical man, ...

**Question:** Where is Mary? **Answer:** hallway

**QA2 two-supporting-facts**

**Facts:** John journeyed to the garden. John grabbed the apple there. Mary travelled to the hallway. Mary went back to the bathroom. Mary went to the garden. Mary travelled to the office. Daniel went to the office. Daniel went to the bedroom. Sandra went back to the office. Sandra journeyed to the garden. Mary travelled to the kitchen. Daniel moved to the kitchen. John put down the apple. Daniel journeyed to the garden. Sandra went to the bathroom. John got the apple there. Daniel travelled to the bedroom. Sandra moved to the hallway. John discarded the apple. Mary travelled to the garden.

**Context:** "From what I have already observed," said Mr. **John journeyed to the garden. John grabbed the apple there. Mary travelled to the hallway. Mary went back to the bathroom.** Ellison, "you will understand that I reject the idea, here expressed, of 'recalling the original beauty of the country.' **Mary went to the garden.** The original beauty is never so great as that which may be introduced. Of course, much depends upon the selection of a spot with capabilities. What is said in respect to the 'detecting and bringing into practice those nice relations of size, proportion and color,' is a mere vagueness of speech, which may mean much, or little, or nothing, and which guides in no degree. **Mary travelled to the office. Daniel went to the office. Daniel went to the bedroom.** That the true 'result of the natural style of gardening is seen rather in the absence of all defects and incongruities, than in the creation of any special wonders or miracles,' is a proposition better suited to the grovelling apprehension of the herd, than to the fervid dreams of the man of genius. **Sandra went back to the office. Sandra journeyed to the garden.** The merit suggested is, at best, negative, and appertains to that hobbling criticism which, in letters, would elevate Addison into apotheosis. **Mary travelled to the kitchen.** In truth, while that merit which consists in the mere avoiding demerit, appeals directly to the understanding, and can thus be foreshadowed in Rule, the loftier merit, which breathes and flames in invention or creation, can be apprehended solely in its results. **Daniel moved to the kitchen. John put down the apple. Daniel journeyed to the garden.** Rule applies but to the excellences of avoidance–to the virtues which deny or refrain. **Sandra went to the bathroom. John got the apple there. Daniel travelled to the bedroom.** We may be instructed to build an Odyssey, but it is in vain that we are told how to conceive a 'Tempest,' an 'Inferno,' a 'Prometheus Bound,' a 'Nightingale,' such as that of Keats, or the 'Sensitive Plant' of Shelley. **Sandra moved to the hallway. John discarded the apple.** But, the thing ...**Mary travelled to the garden.**

**Question:** Where is the apple? **Answer:** garden

**QA3 three-supporting-facts**

**Facts:** Sandra travelled to the office. Sandra picked up the football there. Sandra journeyed to the garden. Sandra journeyed to the bathroom.

**Context:** "From what I have already observed," said Mr. **Sandra travelled to the office.** Ellison, "you will understand that I reject the idea, here expressed, of 'recalling the original beauty of the country.' The original beauty is never so great as that which may be introduced. Of course, much depends upon the selection of a spot with capabilities. What is said in respect to the 'detecting and bringing into practice those nice relations of size, proportion and color,' is a mere vagueness of speech, which may mean much, or little, or nothing, and which guides in no degree. That the true 'result of the natural style of gardening is seen rather in the absence of all defects and incongruities, than in the creation of any special wonders or miracles,' is a proposition better suited to the grovelling apprehension of the herd, than to the fervid dreams of the man of genius. **Sandra picked up the football there.** The merit suggested is, at best, negative, and appertains to that hobbling criticism which, in letters, would elevate Addison into apotheosis. In truth, while that merit which consists in the mere avoiding demerit, appeals directly to the understanding, and can thus be foreshadowed in Rule, the loftier merit, which breathes and flames in invention or creation, can be apprehended solely in its results. Rule applies but to the excellences of avoidance–to the virtues which deny or refrain. **Sandra journeyed to the garden.** We may be instructed to build an Odyssey, but it is in vain that we are told how to conceive a 'Tempest,' an 'Inferno,' a 'Prometheus Bound,' a 'Nightingale,' such as that of Keats, or the 'Sensitive Plant' of Shelley. But, the thing done, the wonder accomplished, and the capacity for apprehension becomes universal. **Sandra journeyed to the bathroom.** The sophists of the negative school, who, through inability to create, have scoffed at creation, are now found the loudest in applause. What, in its chrysalis condition of principle, affronted their demure reason, never fails, in its maturity of accomplishment, to extort admiration from their instinct of the beautiful or of the sublime. " ...

**Question:** Where was the football before the bathroom? **Answer:** garden

**QA4 two-arg-relations**

**Facts:** The garden is south of the bathroom. The bedroom is north of the bathroom.

**Context:** 'A mixture of pure art in a garden scene, adds to it a great beauty.' This is just; and the reference to the sense of human interest is equally so. I repeat that the principle here expressed, is incontrovertible; but there may be something even beyond it. There may be an object in full keeping with the principle suggested–an object unattainable by the means ordinarily in possession of mankind, yet which, if attained, would lend a charm to the landscape-garden immeasurably surpassing that which a merely human interest could bestow. **The garden is south of the bathroom.** The true poet possessed of very unusual pecuniary resources, might possibly, while retaining the necessary idea of art or interest or culture, so imbue his designs at once with extent and novelty of Beauty, as to convey the sentiment of spiritual interference. It will be seen that, in bringing about such result, he secures all the advantages of interest or design, while relieving his work of all the harshness and technicality of Art. **The bedroom is north of the bathroom.** In the most rugged of wildernesses–in the most savage of the scenes of pure Nature–there is apparent the art of a Creator; yet is this art apparent only to reflection; in no respect has it the obvious force of a feeling. Now, if we imagine this sense of the Almighty Design to be harmonized in a measurable degree, if we suppose a landscape whose combined strangeness, vastness, definitiveness, and magnificence, shall inspire the idea of culture, or care, or superintendence, on the part of intelligences superior yet akin to humanity–then the sentiment of interest is preserved, while the Art is made to assume the air of an intermediate or secondary Nature–a Nature which is not God, nor an emanation of God, but which still is Nature, in the sense that it is the handiwork of the angels that hover between man and God." It was in devoting his gigantic wealth to the practical embodiment of a vision such as this–in the free exercise in the open air, which resulted from personal direction of his plans–in the continuous and unceasing object which these plans afford–in the contempt of ambition which it enabled him more to feel than to affect ...

**Question:** What is south of the bathroom? **Answer:** garden

**QA5 three-arg-relations**

**Facts:** Fred grabbed the football there. Jeff took the apple there. Jeff dropped the apple. Bill picked up the apple there. Mary travelled to the kitchen. Mary went back to the hallway. Bill went to the garden. Fred travelled to the garden. Bill passed the apple to Fred. Fred left the apple. Fred went back to the hallway. Fred handed the football to Mary.

**Context:** It was evident that the besiegers were in no hurry; that they were living upon the provisions left in the valley; and that it was their intention to reduce the besieged by famine. **Fred grabbed the football there. Jeff took the apple there.** In fact the inhabitants of the Val d'Avon had been able to carry with them only a small quantity of provisions. **Jeff dropped the apple.** We have described the three kinds of porcelain made in Hizen for exportation to Europe, and we have seen that by the middle of the seventeenth century this commerce, in the hands of the Dutch, and to some extent of the Chinese, had already attained large proportions. Before turning to the kilns that sprung up in other parts of Japan during the eighteenth century–of these the origin in every case can be traced back directly or indirectly to the early Hizen factories–we must say a word about some other varieties of porcelain made in the same neighbourhood, but not destined for foreign use. **Bill picked up the apple there.** The village or town of Arita, of which the better-known Imari is the port, lies about fifty miles to the north-east of Nagasaki, and it may almost be regarded as the King-te-chen of Japan. **Mary travelled to the kitchen. Mary went back to the hallway.** The clay and china-stone used there is now brought, for the most part, from the adjacent islands, from Hirado, from Amakusa, and even from the more remote Goto islands. **Bill went to the garden.** By a combination of some of the most important potters of the district, and with the assistance of some wealthy merchants, a company, the Koransha, was formed some twenty-five years ago,[123] and an attempt was made to keep up the quality of the porcelain produced, at least from a technical point of view. **Fred travelled to the garden.** It was certainly time for some such effort to be made, for about that period, just after the Philadelphia Exhibition, the arts of Japan reached perhaps their nadir. **Bill passed the apple to Fred. Fred left the apple.** MIKÔCHI OR HIRADO WARE.–It was with a somewhat similar object that, ... **Fred went back to the hallway. Fred handed the football to Mary.**

**Question:** What did Bill give to Fred? **Answer:** apple

# M  Author Statement

We confirm that we bear all responsibility in case of any violation of rights that may occur during the collection of data or other aspects of this work. We commit to taking appropriate action, such as removing any data found to be in violation.

# N  BABILong Datasheet

We follow recommended Datasheets for Datasets form (Gebru et al., 2021).

## N.1  Motivation

**For what purpose was the dataset created?**  The BABILong benchmark is designed to test language models' ability to reason across facts distributed in extremely long documents. BABILong includes a diverse set of 20 reasoning tasks, including fact chaining, simple induction, deduction, counting, and handling lists/sets. Today, Large language models (LLMs) and neural architectures are continually evolving and achieving remarkable improvements, particularly in their ability to handle longer contexts, but the benchmarks used to evaluate them have not kept pace. For example, current benchmarks such as Longbench (Bai et al., 2023) and L-Eval (An et al., 2023) scale only up to 40,000 tokens, while models are capable of hundreds of thousands and millions of tokens (OpenAI, 2023b; Bulatov et al., 2024; Gu & Dao, 2023; Anthropic, 2024; Reid et al., 2024; Liu et al., 2024a). To bridge this gap, BABILong allows the construction of tasks of almost arbitrary length, in order to adapt them to the evaluation of new, more powerful models in an extensible and controllable way.

**Who created this dataset (e.g., which team, research group) and on behalf of which entity (e.g., company, institution, organization)?**  This work was done in collaboration of AIRI, Neural Networks and Deep Learning Lab at MIPT, and London Institute for Mathematical Sciences.

**What support was needed to make this dataset?**  Refer to the acknowledgments section of the main text.

## N.2  Composition

**What do the instances that comprise the dataset represent (e.g., documents, photos, people, countries)?**  Each sample is a text document, combined from PG-19 books (Rae et al., 2020) and facts and questions from the bAbI dataset (Weston et al., 2016). The facts are about fictional people, places, animals, and items. PG-19 is a collection of books published before 1919.

**How many instances are there in total (of each type, if appropriate)?**  The BABILong dataset is generative, offering an unlimited number of possible instances. The released pre-generated version includes 13,000 samples, divided into 13 context length splits across 10 tasks, and is available on Hugging Face: `https://huggingface.co/datasets/RMT-team/babilong`. An extended version with 60,000 samples, covering five tasks and offering 1,000 samples per split instead of 100, is also available: `https://huggingface.co/datasets/RMT-team/RMT-team/babilong-1k-samples`.

**Does the dataset contain all possible instances or is it a sample (not necessarily random) of instances from a larger set?**  The test set of BABILong combines sentences of books from the PG-19 (Rae et al., 2020) test split with all test samples from bAbI (Weston et al., 2016) tasks. For evaluation set with 100 samples per task and per length, we randomly sampled 100 test samples from full test set. In train split, we use all train samples from bAbi and randomly sampled texts from PG-19 train split.

**What data does each instance consist of?**  Each sample of BABILong dataset consists of unprocessed sentences of bAbI (Weston et al., 2016) sample (including facts, distractor facts and question) mixed between unprocessed sentences of PG-19 (Rae et al., 2020). The question can be added to either the beginning or the end of the resulting text sequence. See Figure 1a from the main text that illustrates composition of samples in BABILong.

**Is there a label or target associated with each instance?** Yes, each sample in the BABILong dataset is assigned a label, which is the answer to the corresponding question.

**Is any information missing from individual instances?** N/A.

**Are relationships between individual instances made explicit (e.g., users' movie ratings, social network links)?** N/A.

**Are there recommended data splits (e.g., training, development/validation, testing)?** We inherit train and test splits from the bAbI (Weston et al., 2016) dataset. Background texts from PG-19 for the training set are randomly sampled. For the test sets, we fix the background texts and provide pre-generated test splits that we recommend using to report results on the BABILong benchmark (see Section A).

**Are there any errors, sources of noise, or redundancies in the dataset?** The BABILong benchmark uses background texts to hide facts in them. Texts from PG-19 (Rae et al., 2020) may contain mentions of the same enities, places or items that are used in facts from bAbI (Weston et al., 2016). Interference between similar facts in the background text and facts from bAbI can make the benchmark more difficult.

**Is the dataset self-contained, or does it link to or otherwise rely on external resources (e.g., websites, tweets, other datasets)?** Train data relies on bAbI (Weston et al., 2016) and PG-19 datasets both of which are available online. Test sets are self-contained and hosted on HuggingFace (see Section A). We provide code to generate train data of arbitrary lengths (see Section A).

**Does the dataset contain data that might be considered confidential (e.g., data that is protected by legal privilege or by doctor-patient confidentiality, data that includes the content of individuals' non-public communications)?** No.

**Does the dataset contain data that, if viewed directly, might be offensive, insulting, threatening, or might otherwise cause anxiety?** We use books from PG-19 (Rae et al., 2020), a collection of Project Gutenberg books published before 1919. While these texts are generally considered classic literature, it is possible that they contain instances of offensive, insulting, or threatening content, or content that might cause anxiety.

**Does the dataset relate to people?** No.

## N.3 Collection

**How was the data associated with each instance acquired?** The data was directly derived from PG-19 (Rae et al., 2020) and bAbI (Weston et al., 2016) by mixing sentences of these two datasets. No validation or verification of sentence sources was conducted.

**Over what timeframe was the data collected?** The BABILong dataset relies on data from PG-19 (Rae et al., 2020) and bAbI (Weston et al., 2016). The PG-19 corpora contains books published before 1919 and it was released in 2019. The bAbI dataset was released in 2015. BABILong was first uploaded on February 16, 2024.

**What mechanisms or procedures were used to collect the data (e.g., hardware apparatus or sensor, manual human curation, software program, software API)?** All data was collected using the software developed in this paper, which is available on GitHub: `https://github.com/booydar/babilong`. The obtained sequence lengths were validated to match the desired values.

**What was the resource cost of collecting the data?** N/A.

**If the dataset is a sample from a larger set, what was the sampling strategy (e.g., deterministic, probabilistic with specific sampling probabilities)?** The data was directly derived from PG-19 (Rae et al., 2020) and bAbI (Weston et al., 2016) by mixing sentences of these two datasets. For

the desired sequence length we sampled sentences from PG-19 and inserted sentences from a bAbI sample in between them with equal probability, i.e. using the uniform distribution.

**Who was involved in the data collection process (e.g., students, crowdworkers, contractors) and how were they compensated (e.g., how much were crowdworkers paid)?**  BABILong dataset was build by authors of this work.

**Were any ethical review processes conducted (e.g., by an institutional review board)?**  N/A.

**Does the dataset relate to people?**  No.

## N.4  Preprocessing / Cleaning / Labeling

**Was any preprocessing/cleaning/labeling of the data done(e.g., discretization or bucketing, tokenization, part-of-speech tagging, SIFT feature extraction, removal of instances, processing of missing values)?**  To combine texts from PG-19 and bAbI we split books from PG-19 on sentences using nltk.PunktSentenceTokenizer().

**Was the "raw" data saved in addition to the preprocessed/cleaned/labeled data (e.g., to support unanticipated future uses)?**  The BABILong dataset uses data from PG-19 (Rae et al., 2020) and bAbI (Weston et al., 2016), both of which are available online independently.

**Is the software used to preprocess/clean/label the instances available?**  Yes, we provide code that generates BABILong data from PG-19 (Rae et al., 2020) and bAbI (Weston et al., 2016) datasets on-the-fly (see Section A).

## N.5  Uses

**Has the dataset been used for any tasks already?**  Yes, we use the BABILong benchmark to evaluate various large language models and methods for long context processing. The results are presented in the main text of the paper and Section D.

**Is there a repository that links to any or all papers or systems that use the dataset?**  Not yet, but we may add this to the README on GitHub `https://github.com/booydar/babilong`. We have also developed and intend to maintain a leaderboard [10] with up-to-date results.

**What (other) tasks could the dataset be used for?**  The dataset can be used for various tasks beyond long-context evaluation, and we do not restrict its usage to a specific set of tasks. Some possible applications include training and/or evaluating multi-hop question-answering systems or retrieval systems, as BABILong contains multiple facts distributed over long texts that need to be combined to get the correct answer. Additionally, BABILong provides information on which facts are relevant, which can be used for supervision or more detailed metrics and analysis of systems.

**Is there anything about the composition of the dataset or the way it was collected and preprocessed/cleaned/labeled that might impact future uses?**  The BABILong dataset uses texts from the PG-19 corpus (Rae et al., 2020), which consists of books published before 1919. This historical focus might limit the applicability of the dataset to modern language usage and contemporary topics, and it might not represent diverse linguistic styles, dialects, or contemporary societal norms. The reasoning tasks embedded within the texts are designed to challenge specific reasoning abilities in LLMs based on the bAbI dataset (Weston et al., 2016). This method ensures controlled testing conditions but may not accurately reflect the type of reasoning required in real-world scenarios. The synthetic nature of the dataset might also limit a model's ability to generalize from this dataset to natural, unstructured data found in practical applications; however, this remains an open question. Nevertheless, this does not limit the usefulness of the dataset as a benchmark. Additionally, the PG-19 dataset can be replaced with other sources of text, such as Wikipedia.

---

[10]BABILong leaderboard: `https://huggingface.co/spaces/RMT-team/babilong`

**Are there tasks for which the dataset should not be used?** We expect that the BABILong would be used to evaluate long-context processing abilities of LLMs and other long-contex processing architectures. However, we do not restrict any other use cases that a aligned with Project Gutenberg policies and Terms of Use[11] and bAbI's Grant of Patent Rights[12].

## N.6 Distribution

**Will the dataset be distributed to third parties outside of the entity (e.g., company, institution, organization) on behalf of which the dataset was created?** Yes, we use HuggingFace Datasets to host evaluation data and GitHub for code (see Section A).

**How will the dataset will be distributed (e.g., tarball on website, API, GitHub)?** We use HuggingFace Datasets to host evaluation data and Croissant metadata, GitHub for code and data generation (see Section A).

**When will the dataset be distributed?** The BABILong dataset is already available (see Section A).

**Will the dataset be distributed under a copyright or other intellectual property (IP) license, and/or under applicable terms of use (ToU)?** Our code is released under Apache 2.0 License. We use data from PG-19 corpora (Rae et al., 2020) (Apache 2.0 License) and bAbI dataset (Weston et al., 2016) (BSD License). See Section A for links and details on licenses.

**Have any third parties imposed IP-based or other restrictions on the data associated with the instances?** We are not aware of it. We use data from PG-19 corpora (Rae et al., 2020) (Apache 2.0 License) and bAbI dataset (Weston et al., 2016) (BSD License). See Section A for links and details on licenses. PG-19 corpora is a collection of free books from Project Gutenberg[13] published before 1919.

**Do any export controls or other regulatory restrictions apply to the dataset or to individual instances?** No.

## N.7 Maintenance

**Who is supporting/hosting/maintaining the dataset?** The authors of the dataset.

**How can the owner/curator/manager of the dataset be contacted (e.g., email address)?** For inquiries, please reach us via email at {yurii.kuratov,bulatov.as}@phystech.edu, mb@lims.ac.uk, through issues on GitHub, or via Discussions on HuggingFace datasets page (see Section A).

**Is there an erratum?** Any updates will be listed on the README page: https://github.com/booydar/babilong.

**Will the dataset be updated (e.g., to correct labeling errors, add new instances, delete instances)?** Any updates will be listed on the README page: https://github.com/booydar/babilong.

**If the dataset relates to people, are there applicable limits on the retention of the data associated with the instances (e.g., were individuals in question told that their data would be retained for a fixed period of time and then deleted)?** N/A.

**Will older versions of the dataset continue to be supported/hosted/maintained?** Any updates will be listed on the README page: https://github.com/booydar/babilong. Older versions will remain accessible via commit history or by request to the authors.

---

[11]https://www.gutenberg.org/policy/
[12]https://github.com/facebookarchive/bAbI-tasks/blob/master/PATENTS.md
[13]https://www.gutenberg.org/

**If others want to extend/augment/build on/contribute to the dataset, is there a mechanism for them to do so?** Yes, contributions can be made via Pull Requests on GitHub and HuggingFace datasets.

