# OpenReview forum: "BABILong: Testing the Limits of LLMs with Long Context Reasoning-in-a-Haystack"
_NeurIPS.cc/2024/Datasets_and_Benchmarks_Track — NeurIPS 2024 Track Datasets and Benchmarks Spotlight_

### Official Review · Reviewer_rWqi · 2024-07-25
**BABILong: Testing the Limits of LLMs with Long Context Reasoning-in-a-Haystack**

**Rating:** 8
**Confidence:** 4
**Correctness:** Yes
**Clarity:** Everything is clear

**Review:**

The paper "BABILong: Testing the Limits of LLMs with Long Context Reasoning-in-a-Haystack" introduces the BABILong benchmark, aimed at evaluating large language models' (LLMs) ability to reason across facts distributed in very long documents. This benchmark includes 20 diverse reasoning tasks such as fact chaining, induction, deduction, counting, and handling lists/sets. The paper highlights that popular LLMs effectively utilize only 10-20% of the context and struggle significantly with increased reasoning complexity. Despite its innovative design and comprehensive evaluation of over 20 recent LLMs, the paper has some limitations. These include a lack of comparison with other benchmarks like LongBench or RULER, inefficiencies in context utilization, and significant performance degradation as task complexity increases. Retrieval-Augmented Generation (RAG) methods show modest performance, and recurrent memory transformers (RMT) face computational and memory constraints, though they handle sequences up to 11 million tokens. To enhance the benchmark, the paper could benefit from comparisons with other benchmarks, improved context utilization methods, mitigation of performance degradation, enhanced RAG methods, optimized RMT computational requirements, and expanded task diversity. Addressing these areas would make BABILong a more valuable tool for evaluating and advancing LLM capabilities in long-context reasoning tasks.

**Strengths:**

1. BABILong is designed to test the ability of LLMs to handle extremely long contexts, which is crucial as model context sizes continue to increase. The benchmark includes a diverse set of 20 tasks, such as fact chaining, simple induction, deduction, counting, and handling lists/sets, making it comprehensive and challenging.

2. The paper evaluates over 20 recent long-input language models with various sizes and architectures on the BABILong benchmark.
It demonstrates that popular LLMs effectively utilize only 10-20% of the context, with performance declining sharply as reasoning complexity increases.

3. The evaluation reveals that current models, including state-of-the-art ones, struggle significantly with long context reasoning, especially when the facts required for reasoning are distributed across a long document. The paper provides insights into the performance of Retrieval-Augmented Generation (RAG) methods and recurrent memory transformers, showcasing their capabilities and limitations.

4. BABILong is scalable and can be extended to any length, supporting the evaluation of new, more powerful models with increased capabilities. The benchmark's design allows for the construction of tasks of almost arbitrary length, making it adaptable to future advancements in LLMs.

5. The BABILong benchmark data and code for evaluation are publicly available, promoting transparency and facilitating further research and development.

**Additional Feedback:**

No additional feedback

**Documentation:**

NA

**Ethics:**

No ethical issues.

**Limitations:**

1. Include a comprehensive comparison with other benchmarks designed for long-context reasoning to better position BABILong within the landscape of existing evaluation frameworks.

2. Develop methods to improve the efficiency of context utilization by LLMs, ensuring that models can leverage the full extent of the provided context.

3. Investigate and develop strategies to mitigate performance degradation as reasoning complexity increases, possibly through advanced training techniques or architectural innovations.

4, Enhance RAG methods to better handle tasks requiring multiple supporting facts and improve their overall accuracy and robustness across different context lengths.

5.Expand the diversity and complexity of tasks included in the BABILong benchmark to provide a more comprehensive evaluation of LLM capabilities, including more real-world scenarios and complex reasoning tasks.

**Opportunities For Improvement:**

1. The paper does not provide a comprehensive comparison with other benchmarks specifically designed for long context reasoning, such as LongBench or RULER. This limits the understanding of how BABILong performs relative to existing benchmarks.

2. Popular LLMs are shown to utilize only a small fraction of the available context (10-20%), indicating inefficiencies in processing and leveraging long contexts. Do you have any observation why is this happening?

3. RAG methods achieve only modest accuracy (60%) on single-fact question answering, regardless of context length, and perform poorly on more complex tasks requiring multiple supporting facts. Did you experiment with Graph RAG?
Recurrent Memory Transformer Constraints:

4. While recurrent memory transformers (RMT) show promise in handling very long sequences (up to 11 million tokens), they are constrained by computational and memory limitations, affecting their practicality for widespread use. Do you have any observation on which architecture future LLM community should focus more?

5. Although the benchmark includes 20 tasks, the focus remains on relatively simplistic reasoning tasks. Expanding the diversity and complexity of tasks could provide a more comprehensive evaluation of model capabilities.

**Relation To Prior Work:**

Yes it is. However additional comparison should be added regarding other benchmarks

**Summary And Contributions:**

The paper "BABILong: Testing the Limits of LLMs with Long Context Reasoning-in-a-Haystack" introduces the BABILong benchmark, designed to evaluate the ability of large language models (LLMs) to reason across facts distributed in extremely long documents. The benchmark includes a diverse set of 20 reasoning tasks and evaluates various models' performance on these tasks, highlighting significant challenges and limitations in current LLM capabilities.

---

> ### Author Rebuttal · Authors · 2024-08-19
>
> We are grateful for the reviewer’s feedback and advice on further improvements. Below we provide our response to the reviewer’s comments.
>
> **Comparison with other long-context benchmarks**
>
> As the reviewer, we also believe that the development of the LLM benchmarking ecosystem will benefit a lot from a careful contrasting of different approaches. At the beginning of our project we had considered a number of well-established long context benchmarks from the pre-LLM era, including LongRangeArena. However, it is not well suited for evaluating the capabilities of LLMs without direct fine-tuning on tasks. A more recent benchmark, LongBench, includes six types of real and synthetic tasks, ranging from summarization and multidoc QA to code completion. The average sample length in LongBench is 6K tokens for English and 13K tokens for Chinese, with a maximum of 40K tokens. Other existing alternatives available at that time have even shorter document lengths. We mention them in the related work Section 4. Since our main goal is to test sequences of 100K+, LongBench, ∞BENCH, and other earlier methods belong to a qualitatively different category and were not included in detailed comparison.
>
> **Inefficiency of current LLMs in long-context utilization**
>
> We have not explored particular models in much detail but variability in performance between models suggest the following conclusions.
> 1. Cost-efficient context extension methods (YaRN, LongChat, LongAlpaca, Llama-7B-32K) significantly hamper performance compared to more expensive full long context fine-tuning and careful multi-stage context extension (Yi, Phi-3).
> 2. Large recurrent models (Mamba-2.8B, RWKV-5/6) are notoriously weak at needle-in-a-haystack problems and in-context learning, compared to transformers. This can be partially attributed to the bottleneck in the memory state, which does not allow to efficiently memorize all the past information in zero- and few-shot settings. However, we find that introducing transformer layers to state-space recurrent models can mitigate this limitation, as shown by Jamba-51B.
> 3. The improved performance of Llama 3.1 (benchmarked after submission) suggests that curriculum training with increasing context length and carefully collected long context data has a positive effect on context usage.
> 4. Dedicated fine-tuning for tasks requiring longer contexts allows models to handle input more uniformly. Thus, adding long tasks to the model pre-training phase should help.
>
> We plan to extend the paper’s text accordingly.
>
> **GraphRAG**
>
> Thank you for the suggestion to evaluate GraphRAG on BABILong. While this method has the potential to show good performance in submitted study we were mainly focused on evaluation of LLM’s input context and included only one retrieval based method as a reference. RAG methods are booming right now with many paper including GraphRAG (Edge et al. 2024), Graphreader (Li et al. 2024a), HOLMES (Panda et al. 2024), HippoRAG (Guti´errez et al. 2024) published recently. In our view, it would be more beneficial for the community to have a separate study which will have wide and systematic coverage of retrieval augmented generative systems.
>
> **Limitations of RMT and suggestions for future architectures**
>
> As any sequential neural architecture, RMT can’t be straightforwardly parallelized, and as a result, inference time and compute scales linearly with a number of segments. In our case, with GPT-2 as a base model, inference time for RMT will be N x (GPT-2 inference time for 512 tokens) with the constant memory consumption independent of the number of segments. It was shown in [1] that RMT makes transformer models much more computationally efficient, thus on sequences with 2,048,000 tokens and segment size of 512, RMT can run OPT-175B with ×29 fewer FLOPs and OPT-135M with ×295 fewer FLOPs. The main issue with RMT is that it requires complicated curriculum learning and hasn't yet adapted to instruction tuning.
>
> Overall, our analysis (Section 3.1) shows that the best-performing models have the transformer architecture and were generally obtained using full-parameter fine-tuning without cost-efficient extension methods. The hybrid state spaces and transformer architectures, such as Jamba, are also a prominent solution for long context tasks, showing reasonable performance on BABILong. Combining RMT with LLMs for instruction following in long-contexts could also be a promising way to build even more efficient models. So these are possible directions for the community to focus on.
>
> [1] Bulatov, Aydar, et al. "Beyond attention: Breaking the limits of transformer context length with recurrent memory." Proceedings of the AAAI Conference on Artificial Intelligence. Vol. 38. No. 16. 2024.
>
> **On tasks diversity, complexity and real world scenarios**
>
> We agree that the bAbI tasks look simple, and we were surprised that all leading LLM models struggle to achieve high scores for them even without distractor text augmentation. This suggests that it might be too early to introduce more challenging problems until sufficient progress would be made on the current ones.
>
> On the other hand, in this study we develop a benchmarking framework that could be easily scaled to more complex and practically oriented scenarios. For instance, we can mix samples of question-answering datasets with background text from the same domain, or with the help of LLMs generate questions to statements which belong to the original documents. We are looking forward to extend future versions of the dataset with tasks that will challenge and probe capabilities of upcoming LLMs.

---

### Official Review · Reviewer_ZPx6 · 2024-07-25
**A comprehensive and scalable benchmark addressing the limitations of current long-context LLM evaluations**

**Rating:** 7
**Confidence:** 4
**Correctness:** Yes
**Clarity:** Yes

**Review:**

Pros:
- Provides a rigorous assessment of LLMs' long-context processing abilities.
- Includes 20 varied reasoning tasks, ensuring a thorough evaluation of different reasoning aspects.
- Capable of evaluating contexts up to 11 million tokens, setting a new standard for long-context benchmarks.
- Data and evaluation code are publicly available, promoting transparency and further research.
- Offers valuable insights into the limitations and capabilities of current LLMs, guiding future improvements.

Cons:
- Lacks detailed analysis of why certain models perform better or worse, limiting actionable insights for improvement.
- Primarily focuses on synthetic and structured tasks, which may not fully represent practical applications in Real-World Application Scenarios.

**Strengths:**

- The development of BABILong fills a crucial gap in evaluating LLMs' ability to reason across extremely long contexts, offering a more rigorous and comprehensive assessment compared to traditional benchmarks.
- By incorporating 20 varied reasoning tasks, the benchmark effectively tests different aspects of reasoning, such as induction, deduction, and counting, enhancing its relevance and applicability to a wide range of NLP research.
- BABILong's design allows for evaluation on contexts up to 11 million tokens, making it adaptable to future advancements in LLM capabilities and ensuring its long-term utility for the research community.

**Additional Feedback:**

None

**Documentation:**

Yes

**Limitations:**

- The benchmark primarily focuses on synthetic tasks, which may not fully capture the challenges faced in practical applications. Including more real-world scenarios would enhance the relevance and applicability of the benchmark.

**Opportunities For Improvement:**

- A more detailed examination of why certain models perform better or worse on specific tasks could provide actionable insights for enhancing LLM capabilities. Understanding the underlying mechanisms of model performance would help in fine-tuning and developing more robust models.
- Introducing additional evaluation metrics that go beyond accuracy, such as efficiency, scalability, and robustness, could offer a more holistic view of model performance. This would help identify areas where models excel or need improvement.

**Relation To Prior Work:**

Yes

**Summary And Contributions:**

The BABILong benchmark evaluates large language models (LLMs) on their ability to handle and reason across extremely long contexts, addressing gaps in traditional benchmarks. It includes 20 diverse reasoning tasks, such as fact chaining, induction, deduction, and counting, embedded within long texts from the PG19 corpora. Evaluating over 20 recent LLMs, the benchmark reveals that popular models utilize only 10-20% of the context effectively, with performance declining as task complexity increases. Recurrent Memory Transformers showed the highest performance, processing lengths up to 11 million tokens. BABILong is extendable to support the evaluation of new models, with publicly available data and code, providing a comprehensive and scalable evaluation framework for long-context language modeling.

---

> ### Author Rebuttal · Authors · 2024-08-19
>
> We thank the reviewer for valuable feedback. Below we discuss limitations and opportunities for improvement suggested.
>
> **On extending the benchmark with real-world scenarios**
>
> We deeply agree with the reviewer that with addition of more real-world oriented tasks  the proposed benchmark will become more valuable. Indeed, this is the primary motivation behind the BABILong, and as a first step, in this study we develop a method that is scalable to more use cases in future versions of the benchmark. For instance, we can mix samples of question-answering datasets with background text from the same domain, or with the help of LLMs generate questions to statements which belong to the original documents.
>
> **Detailed explanation of models performance**
>
> In our study, we tested more than 20 different open-source, open-weights, and API-only models on five different reasoning tasks. Each model has a unique combination of training data, architectural features, implementation, and training schedule. Moreover, there are no publications for many of the models included. All of these factors severely limit the possibility to characterize behavior of individual models. Still, we were able to make some useful observations, which are listed below.
> 1. Cost-efficient context extension methods (YaRN, LongChat, LongAlpaca, Llama-7B-32K) significantly hamper performance compared to more expensive full long context fine-tuning and careful multi-stage context extension (Yi, Phi-3).
> 2. Large recurrent models (Mamba-2.8B, RWKV-5/6) are notoriously weak at needle-in-a-haystack problems and in-context learning, compared to transformers. This can be partially attributed to the bottleneck in the memory state, which does not allow to efficiently memorize all the past information in zero- and few-shot settings. However, we find that introducing transformer layers to state-space recurrent models can mitigate this limitation, as shown by Jamba-51B.
> 3. Zero-shot BABILong performance, i.e., without instructions and in-context examples, is related to the quality of pre-training, and a few-shot performance, i.e., with instructions and in-context examples, is affected by the quality of the supervised fine-tuning phase..
>
> We plan to extend the Results section of the paper in accordance with these conclusions.
>
> In our follow-up work, we additionally investigate the performance of RMT when training on multiple tasks at once as well as robustness to input perturbations and the dependence of quality on the number of tasks in the training set. This work will be openly available soon.
>
> We are also looking forward to more studies by authors of LLMs as the  benchmark will be adopted by the community. To facilitate this we published predictions of models we tested on github: branch - predictions_06_2024, folder - babilong_evals.
>
> **Additional evaluation metrics**
>
> Thank you for this suggestion, efficiency, scalability, and robustness are critically important for applications. As we are planning the next version of the BABILong benchmark, they are definitely candidates for inclusion. Additionally, we are considering including tests for different positioning of supporting facts in the text.

---

### Official Review · Reviewer_Fwae · 2024-07-27

**Rating:** 9
**Confidence:** 4
**Correctness:** Yes
**Clarity:** Yes

**Review:**

This paper introduces a very important synthetic benchmark to evaluate long context reasoning ability. Unlike needle-in-the-hack which presents an over-simplified evaluation for model's long context ability, BABILong introduces 20 different reasoning tasks and requires the model to pick up information hidden in the long context to infer the final result. Though a synthetic benchmark, this benchmark attempts to simulate the actual long context usage in real life, thus presenting a more natural and realistic evaluation benchmark. In addition, a diverse range of models are evaluated, and interesting observations can be drawn, leading to useful suggestions for the next generation of long context LLMs.

**Strengths:**

1. important benchmark incorporating reasoning in long context ability
2. a diverse range of reasoning tasks are incorporated
3. extensive experiments and interesting results

**Additional Feedback:**

N/A

**Documentation:**

Yes

**Limitations:**

Yes

**Opportunities For Improvement:**

good now.

**Relation To Prior Work:**

Yes

**Summary And Contributions:**

This paper introduces the BABILong benchmark to evaluate model's long context ability, designed to test language models’ ability to reason across facts distributed in extremely long documents. A diverse set of reasoning tasks are involved. Experiments have found that popular LLMs effectively use only 10-20% of the context, with performance declining sharply as length and task complexity increase. RAG methods achieve a modest 60% accuracy in answering single-fact questions, regardless of context length. Among other methods, Mamba and Recurrent Memory Transformers (RMT) show the highest performance, with RMT capable of processing lengths up to 11 million tokens.

---

> ### Author Rebuttal · Authors · 2024-08-19
>
> We are grateful for the comprehensive examination and the high appreciation of our work! We believe that BABILong is a powerful tool for analyzing various aspects of long-context reasoning, and will be beneficial to the community.

---

### Decision · Program_Chairs · 2024-09-26

**Decision:**

Accept (Spotlight)

**Comment:**

The paper introduces the BABILong benchmark to evaluate large language models' (LLMs) ability to reason across long contexts.  BABILong features 20 diverse reasoning tasks that reflect real-life scenarios, challenging models to extract hidden information from lengthy texts. The benchmark tested over many LLMs, and showed that they effectively use only 10-20% of the context, with performance declining as task complexity increases. I think reasoning is one of the most important tasks for future LLM evaluation. The benchmark and its variant could provide more insights on the future LLM performance and robustness.